# PREDICTING WHAT YOU ALREADY KNOW HELPS: PROVABLE SELF-SUPERVISED LEARNING

## ABSTRACT

Self-supervised representation learning solves auxiliary prediction tasks (known as pretext tasks), that do not require labeled data, to learn semantic representations. These pretext tasks are created solely using the input features, such as predicting a missing image patch, recovering the color channels of an image from context, or predicting missing words in text, yet predicting this *known* information helps in learning representations effective for downstream prediction tasks. This paper posits a mechanism based on approximate conditional independence to formalize how solving certain pretext tasks can learn representations that provably decrease the sample complexity of downstream supervised tasks. Formally, we quantify how the approximate independence between the components of the pretext task (conditional on the label and latent variables) allows us to learn representations that can solve the downstream task with drastically reduced sample complexity by just training a linear layer on top of the learned representation.

## 1 INTRODUCTION

Self-supervised learning revitalizes machine learning models in computer vision, language modeling, and control problems (see reference therein (Jing & Tian, 2020; Kolesnikov et al., 2019; Devlin et al., 2018; Wang & Gupta, 2015; Jang et al., 2018)). Training a model with auxiliary tasks based only on input features reduces the extensive costs of data collection and semantic annotations for downstream tasks. It is also known to improve the adversarial robustness of models (Hendrycks et al., 2019; Carmon et al., 2019; Chen et al., 2020a). Self-supervised learning creates pseudo labels solely based on input features, and solves auxiliary prediction tasks in a supervised manner (pretext tasks). However, the underlying principles of self-supervised learning are mysterious since it is a-priori unclear why predicting what we already know should help. We thus raise the following question:

*What conceptual connection between pretext and downstream tasks ensures good representations? What is a good way to quantify this?*

As a thought experiment, consider a simple downstream task of classifying desert, forest, and sea images. A meaningful pretext task is to predict the background color of images (known as image colorization (Zhang et al., 2016)). Denote $X_1, X_2, Y$ to be the input image, color channel, and the downstream label respectively. Given knowledge of the label $Y$, one can possibly predict the background $X_2$ without knowing much about $X_1$. In other words, $X_2$ is approximately independent of $X_1$ conditional on the label $Y$. Consider another task of inpainting (Pathak et al., 2016) the front of a building ($X_2$) from the rest ($X_1$). While knowing the label "building" ($Y$) is not sufficient for successful inpainting, adding additional latent variables $Z$ such as architectural style, location, window positions, etc. will ensure that variation in $X_2$ given $Y, Z$ is small. We can mathematically interpret this as $X_1$ being approximate conditionally independent of $X_2$ given $Y, Z$.

In the above settings with conditional independence, the only way to solve the pretext task for $X_1$ is to first implicitly predict $Y$ and then predict $X_2$ from $Y$. Even without labeled data, the information of $Y$ is hidden in the prediction for $X_2$.

**Contributions.** We propose a mechanism based on approximate conditional independence (ACI) to explain why solving pretext tasks created from known information can learn representations that provably reduce downstream sample complexity. For instance, learned representation will

only require $\tilde{\mathcal{O}}(k)$ samples to solve a $k$-way supervised task under conditional independence (CI). Under ACI (quantified by the norm of a certain partial covariance matrix), we show similar sample complexity improvements. We verify our main Theorem (4.2) using simulations. We check that pretext task helps when CI is approximately satisfied in text domain, and demonstrate on a real-world image dataset that a pretext task-based linear model outperforms or is comparable to many baselines.

## 1.1 RELATED WORK

**Self-supervised learning (SSL) methods in practice:** There has been a flurry of self-supervised methods lately. One class of methods reconstruct images from corrupted or incomplete versions of it, like denoising auto-encoders (Vincent et al., 2008), image inpainting (Pathak et al., 2016), and split-brain autoencoder (Zhang et al., 2017). Pretext tasks are also created using visual common sense, including predicting rotation angle (Gidaris et al., 2018), relative patch position (Doersch et al., 2015), recovering color channels (Zhang et al., 2016), solving jigsaw puzzle games (Noroozi & Favaro, 2016), and discriminating images created from distortion (Dosovitskiy et al., 2015). We refer to the above procedures as reconstruction-based SSL. Another popular paradigm is contrastive learning (Chen et al., 2020b;c). The idea is to learn representations that bring similar data points closer while pushing randomly selected points further away (Wang & Gupta, 2015; Logeswaran & Lee, 2018; Arora et al., 2019) or to maximize a contrastive-based mutual information lower bound between different views (Hjelm et al., 2018; Oord et al., 2018; Tian et al., 2019). A popular approach for text domain is based on language modeling where models like BERT and GPT create auxiliary tasks for next word predictions (Devlin et al., 2018; Radford et al., 2018). The natural ordering or topology of data is also exploited in video-based (Wei et al., 2018; Misra et al., 2016; Fernando et al., 2017), graph-based (Yang et al., 2020; Hu et al., 2019) or map-based (Zhang et al., 2019) self-supervised learning. For instance, the pretext task is to determine the correct temporal order for video frames as in (Misra et al., 2016).

**Theory for self-supervised learning:** Our work initiates some theoretical understanding on the reconstruction-based SSL. Related to our work is the recent theoretical analysis of contrastive learning. Arora et al. (2019) shows guarantees for representations from contrastive learning on *linear classification* tasks using a class conditional independence assumption, but do not handle approximate conditional independence. Recently, Tosh et al. (2020a) show that contrastive learning representations can *linearly* recover any continuous functions of the underlying topic posterior under a topic modeling assumption for text. While their assumption bears some similarity to ours, the assumption of independent sampling of words that they exploit is strong and not generalizable to other domains like images. More recently, concurrent work by Tosh et al. (2020b) shows guarantees for contrastive learning, but not reconstruction-based SSL, with a multi-view redundancy assumptions that is very similar to our CI assumption. (Wang & Isola, 2020) theoretically studies contrastive learning on the hypersphere through intuitive properties like alignment and uniformity of representations; however there is no theoretical connection made to downstream tasks. There is a mutual information maximization view of contrastive learning, but (Tschannen et al., 2019) points out issues with it. Previous attempts to explain negative sampling (Mikolov et al., 2013) based methods use the theory of noise contrastive estimation (Gutmann & Hyvärinen, 2010; Ma & Collins, 2018). However, guarantees are only asymptotic and not for downstream tasks. CI is also used in sufficient dimension reduction Fukumizu et al. (2009; 2004). CI and redundancy assumptions on multiple views (Kakade & Foster, 2007; Ando & Zhang, 2007) are used to analyze a canonical-correlation based dimension reduction algorithm. Finally, (Alain & Bengio, 2014; Vincent, 2011) provide a theoretical analysis for denoising auto-encoder.

## 1.2 OVERVIEW OF RESULTS:

Section 2 introduces notation, setup, and the self-supervised learning procedure considered in this work. In Section 3, we analyze downstream sample complexity under CI. Section 4 presents our main result with relaxed conditions: under ACI with latent variables, and assuming finite samples in both pretext and downstream tasks, for various function classes, and both regression and classification tasks. Experiments verifying our theoretical findings are in Section 5.

## 2 PRELIMINARY

### 2.1 NOTATION

We use lower case symbols ($x$) to denote scalar quantities, bold lower case symbols ($\boldsymbol{x}$) for vector values, capital letters ($X$) for random variables, and capital and bold letters $\boldsymbol{X}$ for matrices. $P_X$ denotes the probability law of random variable $X$, and the space of square-integrable functions with probability $P$ is denoted by $L^2(P)$. We use standard $\mathcal{O}$ notation to hide universal factors and $\tilde{\mathcal{O}}$ to hide log factors. $\| \cdot \|$ stands for $\ell_2$-norm for vectors or Frobenius norm for matrices.

**Linear conditional expectation.** $\mathbb{E}^L[Y|X]$ denotes the prediction of $Y$ with linear regression:

$$\mathbb{E}^L[Y|X = \boldsymbol{x}] := \boldsymbol{W}^*\boldsymbol{x} + \boldsymbol{b}^*, \quad \text{where } \boldsymbol{W}^*, \boldsymbol{b}^* := \arg\min_{\boldsymbol{W}, \boldsymbol{b}} \mathbb{E}[\|Y - \boldsymbol{W}X - \boldsymbol{b}\|^2].$$

In other words, $\mathbb{E}^L[Y|X]$ denotes the best linear predictor of $Y$ given $X$. We also note that $\mathbb{E}[Y|X] \equiv \min_f \mathbb{E}[\|Y - f(X)\|^2]$ is the best predictor of $Y$ given $X$.

**(Partial) covariance matrix.** For random variables $X, Y$, we denote $\boldsymbol{\Sigma}_{XY}$ to be covariance matrix of $X$ and $Y$. For simplicity in most cases, we assume $\mathbb{E}[X] = 0$ and $\mathbb{E}[Y] = 0$; thus we do not distinguish $\mathbb{E}[XY]$ and $\boldsymbol{\Sigma}_{XY}$. The partial covariance matrix between $X$ and $Y$ given $Z$ is:

$$\boldsymbol{\Sigma}_{XY|Z} := \text{cov}\{X - \mathbb{E}^L[X|Z], Y - \mathbb{E}^L[Y|Z]\} \equiv \boldsymbol{\Sigma}_{XY} - \boldsymbol{\Sigma}_{XZ}\boldsymbol{\Sigma}_{ZZ}^{-1}\boldsymbol{\Sigma}_{ZY} \tag{1}$$

**Sub-gaussian random vectors.** A random vector $X \in \mathbb{R}^d$ is $\rho^2$-sub-gaussian if for every fixed unit vector $\boldsymbol{v} \in \mathbb{R}^d$, the variable $\boldsymbol{v}^\top X$ is $\rho^2$-sub-gaussian, i.e., $\mathbb{E}[e^{s\cdot\boldsymbol{v}^\top(X-\mathbb{E}[X])}] \leq e^{s^2\rho^2/2}$ ($\forall s \in \mathbb{R}$).

### 2.2 SETUP AND METHODOLOGY

We denote by $X_1$ the input variable, $X_2$ the target random variable for the pretext task, and $Y$ the label for the downstream task, with $X_1 \in \mathcal{X}_1 \subset \mathbb{R}^{d_1}$, $X_2 \in \mathcal{X}_2 \subset \mathbb{R}^{d_2}$ and $Y \in \mathcal{Y} \subset \mathbb{R}^k$. If $\mathcal{Y}$ is finite with $|\mathcal{Y}| = k$, we assume $\mathcal{Y} \subset \mathbb{R}^k$ is the one-hot encoding of the labels. $P_{X_1X_2Y}$ denotes the joint distribution over $\mathcal{X}_1 \times \mathcal{X}_2 \times \mathcal{Y}$. $P_{X_1Y}, P_{X_1}$ denote the corresponding marginal distributions. Our proposed self-supervised learning procedure is as follows:

*Step 1 (pretext task):* Learn representation $\psi(\boldsymbol{x}_1)$ through $\psi := \arg\min_{g \in \mathcal{H}} \mathbb{E}\|X_2 - g(X_1)\|_F^2$, where $\mathcal{H}$ can be different for different settings that we will specify and discuss later.

*Step 2 (downstream task):* Perform linear regression on $Y$ with $\psi(X_1)$, i.e. $f(\boldsymbol{x}_1) := (\boldsymbol{W}^*)^\top\psi(\boldsymbol{x}_1)$, where $\boldsymbol{W}^* \leftarrow \arg\min_{\boldsymbol{W}} \mathbb{E}_{X_1,Y}[\|Y - \boldsymbol{W}^\top\psi(X_1)\|^2]$. Namely we learn $f(\cdot) = \mathbb{E}^L[Y|\psi(\cdot)]$.

Performance of the learned representation on the downstream task depends on the following quantities. **Approximation error.** We measure this for a learned representation $\psi$ by learning a linear function on top of it for the downstream task. Denote $e_{\text{apx}}(\psi) = \min_{\boldsymbol{W}} \mathbb{E}[\|f^*(X_1) - \boldsymbol{W}\psi(X_1)\|^2]$, where $f^*(\boldsymbol{x}_1) = \mathbb{E}[Y|X_1 = \boldsymbol{x}_1]$ is the optimal predictor for the task. This gives a measure of how well $\psi$ can do with when given infinite samples for the task.

**Estimation error.** We measure sample complexity of $\psi$ on the downstream task and assume access to $n_2$ i.i.d. samples $(\boldsymbol{x}_1^{(1)}, \boldsymbol{y}^{(1)}), \cdots, (\boldsymbol{x}_1^{(n_2)}, \boldsymbol{y}^{(n_2)})$ drawn from $P_{X_1Y}$. We express the $n_2$ samples collectively as $\boldsymbol{X}_1^{\text{down}} \in \mathbb{R}^{n_2 \times d_1}$, $\boldsymbol{Y} \in \mathbb{R}^{n_2 \times k}$ and overload notation to say $\psi(\boldsymbol{X}_1^{\text{down}}) = [\psi(\boldsymbol{x}_1^{(1)})|\psi(\boldsymbol{x}_1^{(2)})\cdots|\psi(\boldsymbol{x}_1^{(n_2)})]^\top \in \mathbb{R}^{n_2 \times d_2}$. We perform linear regression on the learned representation $\psi$ and are interested in the excess risk that measures generalization.

$$\hat{\boldsymbol{W}} \leftarrow \arg\min_{\boldsymbol{W}} \frac{1}{2n_2}\|\boldsymbol{Y} - \psi(\boldsymbol{X}_1)\boldsymbol{W}\|_F^2; \quad \text{ER}_\psi(\hat{\boldsymbol{W}}) := \frac{1}{2}\mathbb{E}\|f^*(X_1) - \hat{\boldsymbol{W}}^\top\psi(X_1)\|_2^2$$

## 3 GUARANTEED RECOVERY WITH CONDITIONAL INDEPENDENCE

In this section, we focus on the case when input $X_1$ and pretext target $X_2$ are conditionally independent (CI) given the downstream label $Y$. While this is a strong assumption that is rarely satisfied in practice, it helps us understand the role of CI with clean results and builds up to our main results with

ACI with latent variables in Section 4. As a warm-up, we show how CI helps when $(X_1, X_2, Y)$ are jointly Gaussian to give us a flavor for the results to follow. We then analyze it for general random variables under two settings: (a) when the function class used for $\psi$ is universal, (b) when $\psi$ is restricted to be a linear function of given features. For now we assume access to a large amount of unlabeled data so as to learn the optimal $\psi^*$ perfectly and this will be relaxed later in Section 4. The general recipe for the results is as follows:

1. Find a closed-form expression for the optimal solution $\psi^*$ for the pretext task.
2. Use conditional independence to argue that $e_{\text{apx}}(\psi^*)$ is small.
3. Exploit the low rank structure of $\psi^*$ to show small estimation error on downstream tasks.

**Data assumption.** Suppose $Y = f^*(X_1) + N$, where $f^* = \mathbb{E}[Y|X_1]$ and hence $\mathbb{E}[N] = 0$. We assume $N$ is $\sigma^2$-subgaussian. For simplicity, we assume non-degeneracy: $\boldsymbol{\Sigma}_{X_i X_i}$, $\boldsymbol{\Sigma}_{YY}$ are full rank.

### 3.1 WARM-UP: JOINTLY GAUSSIAN VARIABLES

We assume $X_1, X_2, Y$ are jointly Gaussian, and so the optimal regression functions are all linear, i.e., $\mathbb{E}[Y|X_1] = \mathbb{E}^L[Y|X_1]$. We also assume data is centered: $\mathbb{E}[X_i] = 0$ and $\mathbb{E}[Y] = 0$. Non-centered data can easily be handled by learning an intercept. All relationships between random variables can then be captured by the (partial) covariance matrix. Therefore it is easy to quantify the CI property and establish the necessary and sufficient conditions that make $X_2$ a reasonable pretext task.

**Assumption 3.1.** *(Jointly Gaussian) $X_1, X_2, Y$ are jointly Gaussian.*

**Assumption 3.2.** *(Conditional independence) $X_1 \perp X_2 | Y$.*

**Claim 3.1** (Closed-form solution). *Under Assumption 3.1, the representation function and optimal prediction that minimize the population risk can be expressed as follows:*

$$\psi^*(\boldsymbol{x}_1) := \mathbb{E}^L[X_2|X_1 = \boldsymbol{x}_1] = \boldsymbol{\Sigma}_{X_2 X_1} \boldsymbol{\Sigma}_{X_1 X_1}^{-1} \boldsymbol{x}_1 \tag{2}$$

$$\textit{Our target } f^*(\boldsymbol{x}_1) := \mathbb{E}^L[Y|X_1 = \boldsymbol{x}_1] = \boldsymbol{\Sigma}_{YX_1} \boldsymbol{\Sigma}_{X_1 X_1}^{-1} \boldsymbol{x}_1. \tag{3}$$

Our prediction for downstream task with representation $\psi^*$ will be: $g(\cdot) := \mathbb{E}^L[Y|\psi^*(X_1)]$. Recall from Equation 1 that the partial covariance matrix between $X_1$ and $X_2$ given $Y$ is $\boldsymbol{\Sigma}_{X_1 X_2 | Y} \equiv \boldsymbol{\Sigma}_{X_1 X_2} - \boldsymbol{\Sigma}_{X_1 Y} \boldsymbol{\Sigma}_{YY}^{-1} \boldsymbol{\Sigma}_{YX_2}$. This partial covariance matrix captures the correlation between $X_1$ and $X_2$ given $Y$. For jointly Gaussian random variables, CI is equivalent to $\boldsymbol{\Sigma}_{X_1 X_2 | Y} = 0$. We first analyze the approximation error based on the property of this partial covariance matrix.

**Lemma 3.2** (Approximation error). *Under Assumption 3.1, 3.2, if $\boldsymbol{\Sigma}_{X_2 Y}$ has rank $k$, $e_{apx}(\psi^*) = 0$.*

**Remark 3.1.** *$\boldsymbol{\Sigma}_{X_2 Y}$ being full column rank implies that $\mathbb{E}[X_2|Y]$ has rank $k$, i.e., $X_2$ depends on all directions of $Y$ and thus captures all directions of information of $Y$. This is a necessary assumption for $X_2$ to be a reasonable pretext task for predicting $Y$. $e_{apx}(\psi^*) = 0$ means $f^*$ is linear in $\psi^*$. Therefore $\psi^*$ selects $d_2$ out of $d_1$ features that are sufficient to predict $Y$.*

Next we consider the estimation error that characterizes the number of samples needed to learn a prediction function $f(\boldsymbol{x}_1) = \hat{\boldsymbol{W}} \psi^*(\boldsymbol{x}_1)$ that generalizes.

**Theorem 3.3** (Estimation error). *Fix a failure probability $\delta \in (0, 1)$. Under Assumption 3.1,3.2, if $n_2 \gg k + \log(1/\delta)$, excess risk of the learned predictor $\boldsymbol{x}_1 \to \hat{\boldsymbol{W}} \psi^*(\boldsymbol{x}_1)$ on the target task satisfies*

$$\text{ER}_{\psi^*}(\hat{\boldsymbol{W}}) \leq \mathcal{O}\left( \frac{\text{Tr}(\boldsymbol{\Sigma}_{YY|X_1})(k + \log(k/\delta))}{n_2} \right),$$

*with probability at least $1 - \delta$.*

Here $\boldsymbol{\Sigma}_{YY|X_1} \equiv \boldsymbol{\Sigma}_{YY} - \boldsymbol{\Sigma}_{YX_1} \boldsymbol{\Sigma}_{X_1 X_1}^{-1} \boldsymbol{\Sigma}_{X_1 Y}$ captures the noise level and is the covariance matrix of tespeche residual term $Y - f^*(X_1) = Y - \boldsymbol{\Sigma}_{YX_1} \boldsymbol{\Sigma}_{X_1 X_1}^{-1} X_1$. Compared to directly using $X_1$ to predict $Y$, self-supervised learning reduces the sample complexity from $\tilde{\mathcal{O}}(d_1)$ to $\tilde{\mathcal{O}}(k)$. We generalize these results even when only a weaker form of CI holds.

**Assumption 3.3** (Conditional independence given latent variables). *There exists some latent variable $Z \in \mathbb{R}^m$ such that $X_1 \perp X_2 | \bar{Y}$, and $\boldsymbol{\Sigma}_{X_2 \bar{Y}}$ is of rank $k + m$, where $\bar{Y} = [Y, Z]$.*

This assumption lets introduce some reasonable latent variables that capture the information between $X_1$ and $X_2$ apart from $Y$. $\boldsymbol{\Sigma}_{X_2\bar{Y}}$ being full rank says that all directions of $\bar{Y}$ are needed to predict $X_2$, and therefore $Z$ is not redundant. For instance, when $Z = X_1$, the assumption is trivially true but $Z$ is not the minimal latent information we want to add. Note it implicitly requires $d_2 \geq k + m$.

**Corollary 3.4.** *Under Assumption 3.1, 3.3, the approximation error $e_{apx}(\psi^*)$ is 0.*

Under CI with latent variable, we can generalize Theorem 3.3 by replacing $k$ by $k + m$.

### 3.2   GENERAL RANDOM VARIABLES

Next we move on to general setting where the variables need not be Gaussian.

**Assumption 3.4.** *Let $X_1 \in \mathbb{R}^{d_1}, X_2 \in \mathbb{R}^{d_2}$ be random variables from some unknown distribution. Let label $Y \in \mathcal{Y}$ be a discrete random variable with $k = |\mathcal{Y}| < d_2$. We assume conditional independence: $X_1 \perp X_2 | Y$.*

Here $Y$ can be interpreted as the multi-class labels where $k$ is the number of classes. For regression problems, one can think about $Y$ as the discretized values of continuous labels. We do not specify the dimension for $Y$ since $Y$ could be arbitrarily encoded but the results only depend on $k$ and the variance of $Y$ (conditional on the input $X_1$).

**Universal function class.**   Suppose we learn the optimal $\psi^*$ among all measurable functions The optimal function $\psi^*$ in this case is naturally given by conditional expectation: $\psi^*(\boldsymbol{x}_1) = \mathbb{E}[X_2|X_1 = \boldsymbol{x}_1]$. We now show that CI implies that $\psi^*$ is good for downstream tasks, which is not apriori clear.

**Lemma 3.5** (Approximation error). *Suppose random variables $X_1, X_2, Y$ satisfy Assumption 3.4, and matrix $\boldsymbol{A} \in \mathbb{R}^{\mathcal{Y} \times d_2}$ with $\boldsymbol{A}_{y,:} := \mathbb{E}[X_2|Y = \boldsymbol{y}]$ is of rank $k = |\mathcal{Y}|$. Then $e_{apx}(\psi^*) = 0$.*

This tells us that although $f^*$ could be nonlinear in $\boldsymbol{x}_1$, it is guaranteed to be linear in $\psi^*(\boldsymbol{x}_1)$. Note that $Y$ does not have to be linear in $X_2$. We provide this simple example for better understanding:

**Example 3.1.** *Let $Y \in \{-1, 1\}$ be binary labels, and $X_1, X_2$ be $2-$mixture Gaussian random variables with $X_1 \sim \mathcal{N}(Y\boldsymbol{\mu}_1, \mathbf{I}), X_2 \sim \mathcal{N}(Y\boldsymbol{\mu}_2, \mathbf{I})$. In this example, $X_1 \perp X_2 | Y$. Although $f^* = \mathbb{E}[Y|X_2]$ is not linear, $\mathbb{E}[Y|\psi]$ is linear: $\psi(\boldsymbol{x}_1) = P(Y = 1|X_1 = \boldsymbol{x}_1)\boldsymbol{\mu}_2 - P(Y = -1|X_1 = \boldsymbol{x}_1)\boldsymbol{\mu}_2$ and $f^*(\boldsymbol{x}_1) = P(Y = 1|X_1 = \boldsymbol{x}_1) - P(Y = -1|X_1 = \boldsymbol{x}_1) \equiv \boldsymbol{\mu}_2^\top \psi(\boldsymbol{x}_1)/\|\boldsymbol{\mu}_2\|^2$.*

Given that $\psi^*$ is good for downstream, we now care about the sample complexity. We will need to assume that the representation has some nice concentration properties. We make an assumption about the whitened data $\psi^*(X_1)$ to ignore scaling factors.

**Assumption 3.5.** *We assume the whitened feature variable $U := \boldsymbol{\Sigma}_\psi^{-1/2}\psi(X_1)$ is a $\rho^2$-subgaussian random variable, where $\boldsymbol{\Sigma}_\psi = \mathbb{E}[\psi(X_1)\psi(X_1)^\top]$.*

We note that all bounded random variables satisfy sub-gaussian property.

**Theorem 3.6** (General conditional independence). *Fix a failure probability $\delta \in (0, 1)$, under the same assumption as Lemma 3.5 and Assumption 3.5 for $\psi^*$, if additionally $n \gg \rho^4(k + \log(1/\delta))$, then the excess risk of the learned predictor $\boldsymbol{x}_1 \to \hat{\boldsymbol{W}}^\top \psi^*(\boldsymbol{x}_1)$ on the downstream task satsifies:*

$$\mathrm{ER}_{\psi^*}[\hat{\boldsymbol{W}}] \leq \mathcal{O}\left(\frac{k + \log(k/\delta)}{n_2}\sigma^2\right).$$

**Function class induced by feature maps.**   Given feature map $\phi_1 : \mathcal{X}_1 \to \mathbb{R}^{D_1}$, we consider the function class $\mathcal{H}_1 = \{\psi : \mathcal{X}_1 \to \mathbb{R}^{d_2} | \exists \boldsymbol{B} \in \mathbb{R}^{d_2 \times D_1}, \psi(\boldsymbol{x}_1) = \boldsymbol{B}\phi_1(\boldsymbol{x}_1)\}$.

**Claim 3.7** (Closed form solution). *The optimal function in $\mathcal{H}$ is $\psi^*(\boldsymbol{x}_1) = \boldsymbol{\Sigma}_{X_2\phi_1}\boldsymbol{\Sigma}_{\phi_1\phi_1}^{-1}\boldsymbol{x}_1$, where $\boldsymbol{\Sigma}_{X_2\phi_1} := \boldsymbol{\Sigma}_{X_2\phi_1(X_1)}$ and $\boldsymbol{\Sigma}_{\phi_1\phi_1} := \boldsymbol{\Sigma}_{\phi_1(X_1)\phi_1(X_1)}$.*

We again show the benefit of CI, this time only comparing the performance of $\psi^*$ to the original features $\phi_1$. Since $\psi^*$ is linear in $\phi_1$, it cannot have smaller approximation error than $\phi_1$. However CI will ensure that $\psi^*$ has the same approximation error as $\phi_1$ and enjoys much better sample complexity.

**Lemma 3.8** (Approximation error). *If Assumption 3.4 is satisfied, and if the matrix $\boldsymbol{A} \in \mathbb{R}^{\mathcal{Y} \times d_2}$ with $\boldsymbol{A}_{y,:} := \mathbb{E}[X_2 | Y = \boldsymbol{y}]$ is of rank $k = |\mathcal{Y}|$. Then $e_{apx}(\psi^*) = e_{apx}(\phi_1)$.*

We additionally need an assumption on the residual $a(\boldsymbol{x}_1) := \mathbb{E}[Y | X_1 = \boldsymbol{x}_1] - \mathbb{E}^L[Y | \phi_1(\boldsymbol{x}_1)]$.

**Assumption 3.6.** *(Bounded approx. error; Condition 3 in Hsu et al. (2012))) We have almost surely*

$$\|\boldsymbol{\Sigma}_{\phi_1 \phi_1}^{-1/2} \phi_1(X_1) a(X_1)^\top \|_F \le b_0 \sqrt{k}$$

**Theorem 3.9.** *(CI with approximation error) Fix a failure probability $\delta \in (0, 1)$, under the same assumption as Lemma 3.8, Assumption 3.5 for $\psi^*$ and Assumption 3.6, if $n_2 \gg \rho^4(k + \log(1/\delta))$, then the excess risk of the learned predictor $\boldsymbol{x}_1 \to \hat{\boldsymbol{W}}^\top \psi^*(\boldsymbol{x}_1)$ on the downstream task satisfies:*

$$\mathrm{ER}_{\psi^*}[\hat{\boldsymbol{W}}] \le e_{apx}(\phi_1) + \mathcal{O}\left( \frac{k + \log(k/\delta)}{n_2} \sigma^2 \right).$$

Theorem 3.9 is also true with Assumption 3.3 instead of exact CI, if we replace $k$ by $km$. Therefore with SSL, the requirement of labels is reduced from complexity for $\mathcal{H}$ to $\mathcal{O}(k)$ ( or $\mathcal{O}(km)$).

**Remark 3.2.** *We note that since $X_1 \perp X_2 | Y$ ensures $X_1 \perp h(X_2) | Y$ for any deterministic function $h$, we could replace $X_2$ by $h(X_2)$ and all results hold. Therefore we could replace $X_2$ with $h(X_2)$ in our algorithm especially when $d_2 < km$.*

## 4 BEYOND CONDITIONAL INDEPENDENCE

In the previous section, we focused on the case where exact CI is satisfied. A weaker but more practical assumption is that $Y$ captures some portion of the dependence between $X_1$ and $X_2$ but not all. A warm-up result with jointly-Gaussian variables is defered in Appendix C.1, where ACI is quantified by the partial covariance matrix. In the section below, we generalize the result from linear function space to arbitrary function space, and introduce the appropriate quantities to measure ACI.

### 4.1 LEARNABILITY WITH GENERAL FUNCTION SPACE

We state the main result with finite samples for both pretext task and downstream task to achieve good generalization. Let $\boldsymbol{X}_1^{\mathrm{pre}} = [\boldsymbol{x}_1^{(1,\mathrm{pre})}, \cdots, \boldsymbol{x}_1^{(n_1,\mathrm{pre})}]^\top \in \mathbb{R}^{n_1 \times d_1}$ and $\boldsymbol{X}_2 = [\boldsymbol{x}_2^{(1)}, \cdots, \boldsymbol{x}_2^{(n_1)}]^\top \in \mathbb{R}^{n_1 \times d_2}$ be the training data from pretext task, where $(\boldsymbol{x}_1^{(i,\mathrm{pre})}, \boldsymbol{x}_2^{(i)})$ is sampled from $P_{X_1 X_2}$. We consider two types of function spaces: $\mathcal{H} \in \{\mathcal{H}_1, \mathcal{H}_u\}$. Recall $\mathcal{H}_1 = \{\psi : \mathcal{X}_1 \to \mathbb{R}^{d_2} | \exists \boldsymbol{B} \in \mathbb{R}^{d_2 \times D_1}, \psi(\boldsymbol{x}_1) = \boldsymbol{B} \phi_1(\boldsymbol{x}_1)\}$ is induced by feature map $\phi_1 : \mathcal{X}_1 \to \mathbb{R}^{D_1}$. $\mathcal{H}_u$ is a function space with universal approximation power (e.g. deep networks) that ensures $\psi^* = \mathbb{E}[X_2 | X_1] \in \mathcal{H}_u$. We learn a representation from $\mathcal{H}$ by using $n_1$ samples: $\tilde{\psi} := \arg\min_{f \in \mathcal{H}_1^{d_2}} \frac{1}{n_1} \|\boldsymbol{X}_2 - f(\boldsymbol{X}_1^{\mathrm{pre}})\|_F^2$. For downstream tasks we similarly define $\boldsymbol{X}_1^{\mathrm{down}} \in \mathbb{R}^{n_2 \times d_1}$, $\boldsymbol{Y} \in \mathbb{R}^{n_2 \times d_3}$[1], and learn a linear classifier trained on $\tilde{\psi}(\boldsymbol{X}_1^{\mathrm{down}})$:

$$\hat{\boldsymbol{W}} \leftarrow \arg\min_{\boldsymbol{W}} \frac{1}{2n_2} \|\boldsymbol{Y} - \tilde{\psi}(\boldsymbol{X}_1^{\mathrm{down}}) \boldsymbol{W}\|_F^2, \ \mathrm{ER}_{\tilde{\psi}}(\hat{\boldsymbol{W}}) := \mathbb{E}_{X_1} \|f_{\mathcal{H}}^*(X_1) - \hat{\boldsymbol{W}}^\top \tilde{\psi}(X_1)\|_2^2.$$

Here $f_{\mathcal{H}}^* = \mathbb{E}^L[Y | \phi_1(X_1)]$ when $\mathcal{H} = \mathcal{H}_1$ and $f_{\mathcal{H}}^* = f^*$ for $\mathcal{H} = \mathcal{H}_u$.

**Assumption 4.1** (Correlation between $X_2$ and $Y, Z$). *Suppose there exists latent variable $Z \in \mathcal{Z}, |\mathcal{Z}| = m$ that ensures $\boldsymbol{\Sigma}_{\phi_{\bar{y}} X_2}$ is full column rank and $\|\boldsymbol{\Sigma}_{Y \phi_{\bar{y}}} \boldsymbol{\Sigma}_{X_2 \phi_{\bar{y}}}^\dagger \|_2 = 1/\beta$, where $A^\dagger$ is pseudo-inverse, and $\phi_{\bar{y}}$ is the one-hot embedding for $\bar{Y} = [Y, Z]$.*

**Definition 4.1** (Approximate conditional independence with function space $\mathcal{H}$).
*1. For $\mathcal{H} = \mathcal{H}_1$, define $\epsilon_{CI} := \|\boldsymbol{\Sigma}_{\phi_1 \phi_1}^{-1/2} \boldsymbol{\Sigma}_{\phi_1 X_2 | \phi_{\bar{y}}}\|_F$.*
*2. For $\mathcal{H} = \mathcal{H}_u$, define $\epsilon_{CI}^2 := \mathbb{E}_{X_1}[\| \mathbb{E}[X_2 | X_1] - \mathbb{E}_{\bar{Y}}[\mathbb{E}[X_2 | \bar{Y}] | X_1]\|^2]$.*

Exact CI for both cases ensures $\epsilon_{\mathrm{CI}} = 0$. We present a unified analysis in the appendix that shows the $\epsilon_{\mathrm{CI}}$ for the second case is same as the first case, with covariance operators instead of matrices.

---

[1] $d_3 = k$ and $Y \equiv \phi_y(Y)$ (one-hot encoding) refers multi-class classification task, $d_3 = 1$ refers to regression.

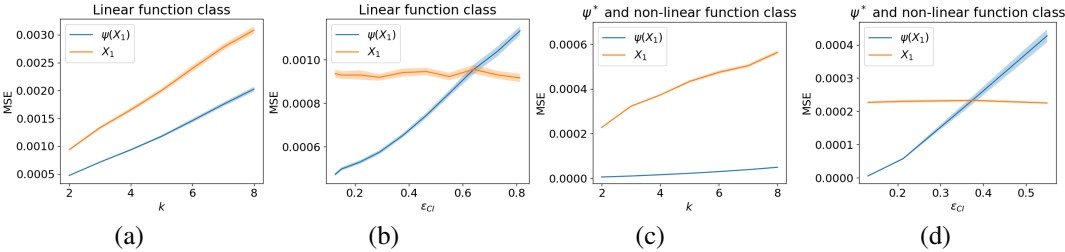

**Figure 1: Left two**: how MSE scales with $k$ (the dimension of $Y$) and $\epsilon_{CI}$ (ACI 4.1) with the linear function class. **Right two**: how MSE scales with $k$ and $\epsilon$ with $\psi^*$ and non-linear function class. Mean of 30 trials are shown in solid line and one standard error is shown by shadow.

When $\mathcal{H} = \mathcal{H}_u$, the residual term $N := Y - \mathbb{E}[Y|X_1]$ is mean zero and assumed to be $\sigma^2$-subgaussian. When we use non-universal features $\phi_1$, $\mathbb{E}[Y - f^*_{\mathcal{H}_1}(X_1)|X_1]$ may not be mean zero. We thus introduce the standard assumption on $a := f^* - f^*_{\mathcal{H}_1} = \mathbb{E}[Y|X_1] - \mathbb{E}^L[Y|\phi_1(X_1)]$:

**Assumption 4.2.** *(Bounded approximation error (Hsu et al., 2012)) There exists a universal constant $b_0$, such that $\|\mathbf{\Sigma}_{\phi_1\phi_1}^{-1/2}\phi_1(X_1)a(X_1)^\top\|_F \leq b_0\sqrt{k}$ almost surely.*

**Theorem 4.2.** *For a fixed $\delta \in (0,1)$, under Assumptions 4.1, 4.2 for $\tilde{\psi}$ and $\psi^*$ and 3.5 for non-universal feature maps, if $n_1, n_2 \gg \rho^4(d_2 + \log 1/\delta)$, and we learn the pretext tasks such that: $\mathbb{E}\|\tilde{\psi}(X_1) - \psi^*(X_1)\|_F^2 \leq \epsilon_{pre}^2$. Then the generalization error for downstream task with probability $1 - \delta$ is:*

$$\mathrm{ER}_{\tilde{\psi}}(\hat{\mathbf{W}}) \leq \mathcal{O}\left(\sigma^2\frac{d_2 + \log(d_2/\delta)}{n_2} + \frac{\epsilon_{CI}^2}{\beta^2} + \frac{\epsilon_{pre}^2}{\beta^2}\right) \tag{4}$$

We defer the proof to the appendix. The proof technique is similar to that of Section 3. The difference is now our $\tilde{\psi}(\mathbf{X}^{(\mathrm{down})}) \in \mathbb{R}^{n_2 \times d_2}$ will be an approximately low rank matrix (low rank + small norm), where the low rank part is the high-signal features that implicitly comes from $Y, Z$ that will be useful for downstream. The remaining part comes from $\epsilon_{CI}$ and $\epsilon_{pre}$. Again by selecting the top $km$ (dimension of $\phi_{\bar{y}}$) features we could further improve the sample complexity:

**Remark 4.1.** *By applying PCA on $\tilde{\psi}(\mathbf{X}_1^{down})$ and keeping the top $km$ principal components only, we can improve the bound in Theorem 4.2 to*

$$\mathrm{ER}_{\tilde{\psi}}(\hat{\mathbf{W}}) \leq \mathcal{O}\left(\sigma^2\frac{km + \log(km/\delta)}{n_2} + \frac{\epsilon_{CI}^2}{\beta^2} + \frac{\epsilon_{pre}^2}{\beta^2}\right). \tag{5}$$

We take a closer look at the different sources of errors in (5): 1) the noise term $Y - f^*(X_1)$ with noise level $\sigma^2$; 2) $\epsilon_{\mathrm{CI}}$ that measures the approximate CI; and 3) $\epsilon_{\mathrm{pre}}$ the error from not learning the pretext task exactly. The first term is optimal setting ignoring log factors as we do linear regression on $mk$-dimensional features. The second and third term are non-reducible due to the fact that $f^*$ is not exactly linear in $\psi$ while we use it as a fixed feature and learn a linear function on it. Therefore it is important to fine-tune when we have sufficient downstream labeled data. We leave this as future work.

Compared to traditional supervised learning, learning $f^*_{\mathcal{H}}$ requires sample complexity scaling with the (Rademacher/Gaussian) complexity of $\mathcal{H}$ (see e.g. Bartlett & Mendelson (2002); Shalev-Shwartz & Ben-David (2014)), which is very large for complicated models such as deep networks.

In Section D, we consider a similar result for cross-entropy loss.

## 5 EXPERIMENTS

In this section, we empirically verify our claim that SSL performs well when ACI is satisfied.

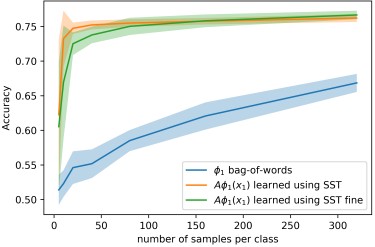 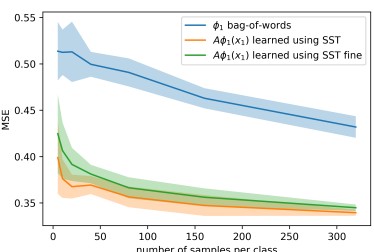

Figure 2: Performance on SST of baseline $\phi_1(\boldsymbol{x}_1)$, i.e. bag-of-words, and learned $\psi(\boldsymbol{x}_1)$ for the two settings. **Left:** Classification accuracy, **Right:** Regression MSE.

**Simulations.** With synthetic data, we verify how excess risk (ER) scales with the cardinality/feature dimension of $\mathcal{Y}$ ($k$), and ACI ($\epsilon_{CI}$ in Definition 4.1). We consider a mixture of Gaussian data and conduct experiments with both linear function space ($\mathcal{H}_1$ with $\phi_1$ as identity map) and universal function space $\mathcal{H}_u$. We sample the label $Y$ uniformly from $\{1, ..., k\}$. For $i$-th class, the centers $\mu_{1i} \in \mathbb{R}^{d_1}$ and $\mu_{2i} \in \mathbb{R}^{d_2}$ are uniformly sampled from $[0, 10]$. Given $Y = i$, $\alpha \in [0, 1]$, let $X_1 \sim \mathcal{N}(\mu_{1i}, \mathbf{I})$, $\hat{X}_2 \sim \mathcal{N}(\mu_{2i}, \mathbf{I})$, and $X_2 = (1 - \alpha)\hat{X}_2 + \alpha X_1$. Therefore $\alpha$ is a correlation coefficient: $\alpha = 0$ ensures $X_2$ being CI with $X_1$ given $Y$ and when $\alpha = 1$, $X_2$ fully depends on $X_1$. (if $d_1 \neq d_2$, we append zeros or truncate to fit accordingly).

We first conduct experiments with linear function class. We learn a linear representation $\psi$ with $n_1$ samples and the linear prediction of $Y$ from $\psi$ with $n_2$ samples. We set $d_1 = 50$, $d_2 = 40$, $n_1 = 4000$, $n_2 = 1000$ and ER is measured with Mean Squared Error (MSE). As shown in Figure 1(a)(b), the MSE of learning with $\psi(X_1)$ scales linearly with $k$ as indicated in Theorem 3.9, and scales linearly with $\epsilon_{CI}$ associated with linear function class as indicated in Theorem 4.2. Next we move on to general function class, i.e., $\psi^* = \mathbb{E}[Y|X_1]$ with a closed form solution (see example 3.1). We use the same parameter settings as above. For baseline method, we use kernel linear regression to predict $Y$ using $X_1$ (we use RBF kernel which also has universal approximation power). As shown in Figure 1(c)(d), the phenomenon is the same as what we observe in the linear function class setting, and hence they respectively verify Theorem 3.6 and Theorem 4.2 with $\mathcal{H}_u$.

**NLP task.** We look at the setting where both $\mathcal{X}_1$ and $\mathcal{X}_2$ are the set of sentences and perform experiments by enforcing CI with and without latent variables. The downstream task is sentiment analysis with the Stanford Sentiment Treebank (SST) dataset (Socher et al., 2013), where inputs are movie reviews and the label set $\mathcal{Y}$ is $\{\pm 1\}$. We use the representation class $\mathcal{H}_1$, with features $\phi_1$ being the bag-of-words representation ($D_1 = 13848$). For $X_2$ we use a $d_2 = 300$ dimensional embedding of the sentence, that is the mean of word vectors (random gaussians) for the words in the sentence. For SSL data we consider 2 settings, (a) enforce CI with the labels $\mathcal{Y}$, (b) enforce CI with extra latent variables, for which we use fine-grained version of SST with label set $\bar{\mathcal{Y}} = \{1, 2, 3, 4, 5\}^2$. We test the learned $\psi$ on SST binary task with linear regression and linear classification; results are presented in Figure 2. We observe that in both settings $\psi$ outperforms $\phi_1$, especially in the small-sample-size regime. Also exact CI is better than CI with extra latent variables, as suggested by theory.

## 6 CONCLUSION

In this work we theoretically quantify how an approximate conditional independence assumption that connects pretext and downstream task data distributions can give sample complexity benefits of self-supervised learning on downstream tasks. Our theoretical findings are also supported by experiments on simulated data and also on real CV and NLP tasks. We would like to note that approximate CI is only a sufficient condition for a useful pretext task. We leave it for future work to investigate other mechanisms by which pretext tasks help with downstream tasks.

---

[2] Ratings $\{1, 2\}$ correspond to $y = -1$ and $\{4, 5\}$ correspond to $y = 1$

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

# A    SOME USEFUL FACTS

## A.1    RELATION OF INVERSE COVARIANCE MATRIX AND PARTIAL CORRELATION

en For a covariance matrix of joint distribution for variables $X, Y$, the covariance matrix is

$$\begin{bmatrix} \boldsymbol{\Sigma}_{XX} & \boldsymbol{\Sigma}_{XY} \\ \boldsymbol{\Sigma}_{YX} & \boldsymbol{\Sigma}_{YY} \end{bmatrix} = \begin{bmatrix} \boldsymbol{\Sigma}_{X_1 X_1} & \boldsymbol{\Sigma}_{X_1 X_2} & \boldsymbol{\Sigma}_{X_1 Y} \\ \boldsymbol{\Sigma}_{X_2 X_1} & \boldsymbol{\Sigma}_{X_2 X_2} & \boldsymbol{\Sigma}_{X_2 Y} \\ \boldsymbol{\Sigma}_{YX_1} & \boldsymbol{\Sigma}_{X_2 Y} & \boldsymbol{\Sigma}_{YY} \end{bmatrix}.$$

Its inverse matrix $\boldsymbol{\Sigma}^{-1}$ satisfies

$$\boldsymbol{\Sigma}^{-1} = \begin{bmatrix} \boldsymbol{A} & \rho \\ \rho^\top & \boldsymbol{B} \end{bmatrix}.$$

Here $\boldsymbol{A}^{-1} = \boldsymbol{\Sigma}_{XX} - \boldsymbol{\Sigma}_{XY}\boldsymbol{\Sigma}_{YY}^{-1}\boldsymbol{\Sigma}_{YX} \equiv \text{cov}(X - \mathbb{E}^L[X|Y], X - \mathbb{E}^L[X|Y]) := \boldsymbol{\Sigma}_{XX \cdot Y}$, the partial covariance matrix of $X$ given $Y$.

## A.2    RELATION TO CONDITIONAL INDEPENDENCE

*Proof of Lemma C.5.*

**Fact A.1.** *When $X_1 \perp X_2 | Y$, the partial covariance between $X_1, X_2$ given $Y$ is 0:*

$$\boldsymbol{\Sigma}_{X_1 X_2 \cdot Y} := \text{cov}(X_1 - \mathbb{E}^L[X_1|Y], X_2 - \mathbb{E}^L[X_2|Y])$$
$$\equiv \boldsymbol{\Sigma}_{X_1 X_2} - \boldsymbol{\Sigma}_{X_1 Y}\boldsymbol{\Sigma}_{YY}^{-1}\boldsymbol{\Sigma}_{YX_2} = 0.$$

The derivation comes from the following:

**Lemma A.1** (Conditional independence (Adapted from Huang (2010))). *For random variables $X_1, X_2$ and a random variable $Y$ with finite values, conditional independence $X_1 \perp X_2 | Y$ is equivalent to:*

$$\sup_{f \in N_1, g \in N_2} \mathbb{E}[f(X_1)g(X_2)|Y] = 0. \tag{6}$$

*Here $N_i = \{f : \mathbb{R}^{d_i} \to R : E[f(X_i)|Y] = 0\}, i = 1, 2.$*

Notice for arbitrary function $f$, $\mathbb{E}[f(X)|Y] = \mathbb{E}^L[f(X)|\phi_y(Y)]$ with one-hot encoding of discrete variable $Y$. Therefore for any feature map we can also get that conditional independence ensures:

$$\boldsymbol{\Sigma}_{\phi_1(X_1)\phi_2(X_2)|Y} := \text{cov}(\phi_1(X_1) - \mathbb{E}^L[\phi_1(X_1)|\phi_y(Y)], \phi_2(X_2) - \mathbb{E}^L[\phi_2(X_2)|\phi_y(Y)])$$
$$= \mathbb{E}[\bar{\phi}_1(X_1)\bar{\phi}_2(X_2)^\top] = 0.$$

Here $\bar{\phi}_1(X_1) = \phi_1(X_1) - \mathbb{E}[\phi_1(X_1)|\phi_y(Y)]$ is mean zero given $Y$, and vice versa for $\bar{\phi}_2(X_2)$. This thus finishes the proof for Lemma C.5. $\qquad\square$

## A.3    TECHNICAL FACTS FOR MATRIX CONCENTRATION

We include this covariance concentration result that is adapted from Claim A.2 in Du et al. (2020):

**Claim A.2** (covariance concentration for gaussian variables). *Let $\boldsymbol{X} = [x_1, x_2, \cdots x_n]^\top \in \mathbb{R}^{n \times d}$ where each $x_i \sim \mathcal{N}(0, \boldsymbol{\Sigma}_X)$. Suppose $n \gg k + \log(1/\delta)$ for $\delta \in (0, 1)$. Then for any given matrix $B \in \mathbb{R}^{d \times m}$ that is of rank $k$ and is independent of $\boldsymbol{X}$, with probability at least $1 - \frac{\delta}{10}$ over $\boldsymbol{X}$ we have*

$$0.9\boldsymbol{B}^\top\boldsymbol{\Sigma}_X\boldsymbol{B} \preceq \frac{1}{n}\boldsymbol{B}^\top\boldsymbol{X}^\top\boldsymbol{X}\boldsymbol{B} \preceq 1.1\boldsymbol{B}^\top\boldsymbol{\Sigma}_X\boldsymbol{B}. \tag{7}$$

And we will also use Claim A.2 from Du et al. (2020) for concentrating subgaussian random variable.

**Claim A.3** (covariance concentration for subgaussian variables). *Let $\boldsymbol{X} = [x_1, x_2, \cdots x_n]^\top \in \mathbb{R}^{n \times d}$ where each $x_i \sim \mathcal{N}(0, \boldsymbol{\Sigma}_X)$. Suppose $n \gg \rho^4(k + \log(1/\delta))$ for $\delta \in (0, 1)$. Then for any given matrix $B \in \mathbb{R}^{d \times m}$ that is of rank $k$ and is independent of $\boldsymbol{X}$, with probability at least $1 - \frac{\delta}{10}$ over $\boldsymbol{X}$ we have*

$$0.9\boldsymbol{B}^\top\boldsymbol{\Sigma}_X\boldsymbol{B} \preceq \frac{1}{n}\boldsymbol{B}^\top\boldsymbol{X}^\top\boldsymbol{X}\boldsymbol{B} \preceq 1.1\boldsymbol{B}^\top\boldsymbol{\Sigma}_X\boldsymbol{B}. \tag{8}$$

**Claim A.4.** *Let $\boldsymbol{Z} \in \mathbb{R}^{n \times k}$ be a matrix with row vectors sampled from i.i.d Gaussian distribution $\mathcal{N}(0, \boldsymbol{\Sigma}_Z)$. Let $P \in \mathbb{R}^{n \times n}$ be a fixed projection onto a space of dimension $d$. Then with a fixed $\delta \in (0, 1)$, we have:*

$$\|P\boldsymbol{Z}\|_F^2 \lesssim \mathrm{Tr}(\boldsymbol{\Sigma}_Z)(d + \log(k/\delta)),$$

*with probability at least $1 - \delta$.*

*Proof of Claim A.4.* Each $t$-th column of $Z$ is an $n$-dim vector that is i.i.d sampled from Gaussian distribution $\mathcal{N}(0, \boldsymbol{\Sigma}_{tt})$.

$$\|P\boldsymbol{Z}\|_F^2 = \sum_{t=1}^{k} \|P\boldsymbol{z}_t\|^2$$

$$= \sum_{t=1}^{k} \boldsymbol{z}_t^\top P \boldsymbol{z}_t.$$

Each term satisfy $\boldsymbol{\Sigma}_{kk}^{-1}\|P\boldsymbol{z}_t\|^2 \sim \chi^2(d)$, and therefore with probability at least $1 - \delta'$ over $\boldsymbol{z}_t$,

$$\boldsymbol{\Sigma}_{kk}^{-1}\|P\boldsymbol{z}_t\|^2 \lesssim d + \log(1/\delta').$$

Using union bound, take $\delta' = \delta/k$ and summing over $t \in [k]$ we get:

$$\|P\boldsymbol{Z}\|_F^2 \lesssim \mathrm{Tr}(\boldsymbol{\Sigma}_Z)(d + \log(k/\delta)).$$

$\square$

**Theorem A.5** (Hanson-Wright Inequality (Theorem 1.1 from Rudelson et al. (2013))). *Let $X = (X_1, X_2, \cdots X_n) \in \mathbb{R}^n$ be a random vector with independent components $X_1$ which satisfy $\mathbb{E}[X_i] = 0$ and $\|X_i\|_{\psi_2} \leq K$. Let $A$ be an $n \times n$ matrix. Then, for every $t \geq 0$,*

$$\mathbb{P}\left[|X^\top A X - \mathbb{E}[X^\top A X]| > t\right] \leq 2\exp\left\{-c\min\left(\frac{t^2}{K^4\|A\|_F^2}, \frac{t}{K^2\|A\|}\right)\right\}.$$

**Theorem A.6** (Vector Bernstein Inequality (Theorem 12 in Gross (2011))). *Let $X_1, \cdots, X_m$ be independent zero-mean vector-valued random variables. Let*

$$N = \|\sum_{i=1}^{m} X_i\|_2.$$

*Then*

$$\mathbb{P}[N \geq \sqrt{V} + t] \leq \exp\left(\frac{-t^2}{4V}\right),$$

*where $V = \sum_i \mathbb{E}\|X_i\|_2^2$ and $t \leq V/(\max\|X_i\|_2)$.*

**Lemma A.7.** *Let $\boldsymbol{Z} \in \mathbb{R}^{n \times k}$ be a matrix whose row vectors are $n$ independent mean-zero (conditional on $P$) $\sigma$-sub-Gaussian random vectors. With probability $1 - \delta$:*

$$\|P\boldsymbol{Z}\|^2 \lesssim \sigma^2(d + \log(d/\delta)).$$

*Proof of Lemma A.7.* Write $P = \boldsymbol{U}\boldsymbol{U}^\top = [\boldsymbol{u}_1, \cdots, \boldsymbol{u}_d]$ where $\boldsymbol{U}$ is orthogonal matrix in $\mathbb{R}^{n \times d}$ where $\boldsymbol{U}^\top \boldsymbol{U} = I$.

$$\|P\boldsymbol{Z}\|_F^2 = \|\boldsymbol{U}^\top \boldsymbol{Z}\|_F^2$$

$$= \sum_{j=1}^{d} \|\boldsymbol{u}_j^\top \boldsymbol{Z}\|^2$$

$$= \sum_{j=1}^{d} \|\sum_{i=1}^{n} \boldsymbol{u}_{ji}\boldsymbol{z}_i\|^2,$$

where each $\boldsymbol{z}_i \in \mathbb{R}^k$ being the $i$-th row of $\boldsymbol{Z}$ is a centered independent $\sigma$ sub-Gaussian random vectors. To use vector Bernstein inequality, we let $X := \sum_{i=1}^n X_i$ with $X_i := \boldsymbol{u}_{ji}\boldsymbol{z}_i$. We have $X_i$ is zero mean: $\mathbb{E}[X_i] = \mathbb{E}[\boldsymbol{u}_{ji}\,\mathbb{E}[\boldsymbol{z}_i|\boldsymbol{u}_{ji}]] = \mathbb{E}[\boldsymbol{u}_{ji} \cdot 0] = 0$.

$$
\begin{aligned}
V := &\sum_i \mathbb{E}\,\|X_i\|_2^2 \\
= &\sum_i \mathbb{E}[\boldsymbol{u}_{ji}^2 \boldsymbol{z}_i^\top \boldsymbol{z}_i] \\
= &\sum_i \mathbb{E}_{\boldsymbol{u}_{ji}}[\boldsymbol{u}_{ji}^2\,\mathbb{E}[\|\boldsymbol{z}_i\|_2^2|\boldsymbol{u}_{ji}]] \\
\leq &\sigma^2 \sum_i \mathbb{E}_{\boldsymbol{u}_{ji}}[\boldsymbol{u}_{ji}^2] \\
= &\sigma^2.
\end{aligned}
$$

Therefore by vector Bernstein Inequality, with probability at least $1 - \delta/d$, $\|X\| \leq \sigma(1 + \sqrt{\log(d/\delta)})$. Then by taking union bound, we get that $\|P\boldsymbol{Z}\|^2 = \sum_{j=1}^d \|\boldsymbol{u}_j^\top \boldsymbol{Z}\|^2 \lesssim \sigma^2(d + \log(d/\delta))$ with probability $1 - \delta$.

$\square$

**Corollary A.8.** *Let $\boldsymbol{Z} \in \mathbb{R}^{n \times k}$ be a matrix whose row vectors are $n$ independent samples from centered (conditioned on $P$) multinomial probabilities $(p_1, p_2, \cdots p_k)$ (where $p_t$ could be different across each row). Let $P \in \mathbb{R}^{n \times n}$ be a projection onto a space of dimension $d$ (that might be dependent with $\boldsymbol{Z}$). Then we have*

$$
\|P\boldsymbol{Z}\|^2 \lesssim d + \log(d/\delta).
$$

*with probability $1 - \delta$.*

# B  OMITTED PROOFS WITH CONDITIONAL INDEPENDENCE

*Proof of Lemma 3.2.*

$$
\text{cov}(X_1|Y, X_2|Y) = \boldsymbol{\Sigma}_{X_1 X_2} - \boldsymbol{\Sigma}_{X_1 Y}\boldsymbol{\Sigma}_{YY}^{-1}\boldsymbol{\Sigma}_{Y X_2} = 0.
$$

By plugging it into the expression of $\mathbb{E}^L[X_2|X_1]$, we get that

$$
\begin{aligned}
\psi(x_1) := \mathbb{E}^L[X_2|X_1 = x_1] &= \boldsymbol{\Sigma}_{X_2 X_1}\boldsymbol{\Sigma}_{X_1 X_1}^{-1} x_1 \\
&= \boldsymbol{\Sigma}_{X_2 Y}\boldsymbol{\Sigma}_{YY}^{-1}\boldsymbol{\Sigma}_{Y X_1}\boldsymbol{\Sigma}_{X_1 X_1}^{-1} x_1 \\
&= \boldsymbol{\Sigma}_{X_2 Y}\boldsymbol{\Sigma}_{YY}^{-1}\,\mathbb{E}^L[Y|X_1].
\end{aligned}
$$

Therefore, as long as $\boldsymbol{\Sigma}_{X_2 Y}$ of rank $k$, it has left inverse matrix and we get: $\mathbb{E}^L[Y|X_1 = x_1] = \boldsymbol{\Sigma}_{X_2 Y}^\dagger \boldsymbol{\Sigma}_{YY} \psi(x_1)$. Therefore there's no approximation error in using $\psi$ to predict $Y$.

$\square$

*Proof of Corollary 3.4 .* Let selector operator $S_y$ be the mapping such that $S_y \bar{Y} = Y$, we overload it as the matrix that ensure $S_y \boldsymbol{\Sigma}_{\bar{Y} X} = \boldsymbol{\Sigma}_{Y X}$ for any random variable $X$ as well.

From Lemma 3.2 we get that there exists $W$ such that $\mathbb{E}^L[\bar{Y}|X_1] = W\,\mathbb{E}^L[X_2|X_1]$, just plugging in $S_y$ we get that $\mathbb{E}^L[Y|X_1] = (S_y W)\,\mathbb{E}^L[X_2|X_1]$.

$\square$

*Proof of Theorem 3.3 .* Since $N$ is mean zero, $f^*(X_1) = \mathbb{E}[Y|X_1] = (\boldsymbol{A}^*)^\top X_1$.

$\mathbb{E}^L[Y|X_1 = x_1] = \boldsymbol{\Sigma}_{X_2 Y}^\dagger \boldsymbol{\Sigma}_{YY} \psi(x_1)$. Let $\boldsymbol{W}^* = \boldsymbol{\Sigma}_{YY}\boldsymbol{\Sigma}_{Y X_2}^\dagger$.

First we have the basic inequality,

$$\frac{1}{2n_2}\|\boldsymbol{Y} - \psi(\boldsymbol{X}_1)\hat{\boldsymbol{W}}\|_F^2 \leq \frac{1}{2n_2}\|\boldsymbol{Y} - \boldsymbol{X}_1 A^*\|_F^2$$

$$= \frac{1}{2n_2}\|\boldsymbol{Y} - \psi(\boldsymbol{X}_1)\boldsymbol{W}^*\|_F^2.$$

Therefore

$$\|\psi(\boldsymbol{X}_1)\boldsymbol{W}^* - \psi(\boldsymbol{X}_1)\hat{\boldsymbol{W}}\|^2 \leq 2\langle N, \psi(\boldsymbol{X}_1)\boldsymbol{W}^* - \psi(\boldsymbol{X}_1)\hat{\boldsymbol{W}}\rangle$$

$$= 2\langle P_{\psi(\boldsymbol{X}_1)}\boldsymbol{N}, \psi(\boldsymbol{X}_1)\boldsymbol{W}^* - \psi(\boldsymbol{X}_1)\hat{W}\rangle$$

$$\leq 2\|P_{\psi(\boldsymbol{X}_1)}\boldsymbol{N}\|_F\|\psi(\boldsymbol{X}_1)\boldsymbol{W}^* - \psi(\boldsymbol{X}_1)\hat{W}\|_F$$

$$\Rightarrow \|\psi(\boldsymbol{X}_1)\boldsymbol{W}^* - \psi(\boldsymbol{X}_1)\hat{\boldsymbol{W}}\| \leq 2\|P_{\psi(\boldsymbol{X}_1)}\boldsymbol{N}\|_F$$

$$\lesssim \sqrt{\mathrm{Tr}(\boldsymbol{\Sigma}_{YY|X_1})(k + \log k/\delta)}. \qquad \text{(from Claim A.4)}$$

The last inequality is derived from Claim A.7 and the fact that each row of $\boldsymbol{N}$ follows gaussian distribution $\mathcal{N}(0, \boldsymbol{\Sigma}_{YY|X_1})$. Therefore

$$\frac{1}{n_2}\|\psi(\boldsymbol{X}_1)W^* - \psi(\boldsymbol{X}_1)\hat{W}\|_F^2 \lesssim \frac{\mathrm{Tr}(\boldsymbol{\Sigma}_{YY|X_1})(k + \log k/\delta)}{n_2}.$$

Next we need to concentrate $1/n\boldsymbol{X}_1^\top \boldsymbol{X}_1$ to $\boldsymbol{\Sigma}_X$. Suppose $\mathbb{E}^L[X_2|X_1] = \boldsymbol{B}^\top X_1$, i.e., $\phi(x_1) = \boldsymbol{B}^\top x_1$, and $\phi(\boldsymbol{X}_1) = \boldsymbol{X}_1\boldsymbol{B}$. With Claim A.2 we have $1/n\phi(\boldsymbol{X}_1)^\top \phi(\boldsymbol{X}_1) = 1/n\boldsymbol{B}^\top \boldsymbol{X}_1^\top \boldsymbol{X}_1 \boldsymbol{B}$ satisfies:

$$0.9\boldsymbol{B}^\top\boldsymbol{\Sigma}_X\boldsymbol{B} \preceq 1/n_2\phi(\boldsymbol{X}_1)^\top\phi(\boldsymbol{X}_1) \preceq 1.1\boldsymbol{B}^\top\boldsymbol{\Sigma}_X\boldsymbol{B}$$

Therefore we also have:

$$\mathbb{E}[(\boldsymbol{W}^* - \hat{\boldsymbol{W}})^\top\psi(x_1)]$$

$$= \|\boldsymbol{\Sigma}_X^{1/2}\boldsymbol{B}(\boldsymbol{W}^* - \hat{\boldsymbol{W}})\|_F^2$$

$$\leq \frac{1}{0.9n_2 k}\|\psi(\boldsymbol{X}_1)W^* - \psi(\boldsymbol{X}_1)\hat{W}\|_F^2 \lesssim \frac{\mathrm{Tr}(\boldsymbol{\Sigma}_{YY|X_1})(k + \log k/\delta)}{n_2}.$$

$\square$

## B.1 OMITTED PROOF FOR GENERAL RANDOM VARIABLES

*Proof of Lemma 3.5.* Let the representation function $\psi$ be defined as:

$$\psi(\cdot) := \mathbb{E}[X_2|X_1] = \mathbb{E}[\mathbb{E}[X_2|X_1, Y]|X_1]$$

$$= \mathbb{E}[\mathbb{E}[X_2|Y]|X_1] \qquad \text{(uses CI)}$$

$$= \sum_y P(Y = y|X_1)\,\mathbb{E}[X_2|Y = y]$$

$$=: f(X_1)^\top \boldsymbol{A},$$

where $f : \mathbb{R}^{d_1} \to \Delta_{\mathcal{Y}}$ satisfies $f(x_1)_y = P(Y = y|X_1 = x_1)$, and $\boldsymbol{A} \in \mathbb{R}^{\mathcal{Y} \times d_2}$ satisfies $\boldsymbol{A}_{y,:} = \mathbb{E}[X_2|Y = y]$. Here $\Delta_d$ denotes simplex of dimension $d$, which represents the discrete probability density over support of size $d$.

Let $\boldsymbol{B} = \boldsymbol{A}^\dagger \in \mathbb{R}^{\mathcal{Y} \times d_2}$ be the pseudoinverse of matrix $A$, and we get $\boldsymbol{B}\boldsymbol{A} = \boldsymbol{I}$ from our assumption that $\boldsymbol{A}$ is of rank $|\mathcal{Y}|$. Therefore $f(x_1) = \boldsymbol{B}\psi(x_1), \forall x_1$. Next we have:

$$\mathbb{E}[Y|X_1 = x_1] = \sum_y P(Y = y|X_1 = x_1) \times y$$

$$= \boldsymbol{Y}f(x_1)$$

$$= (\boldsymbol{Y}\boldsymbol{B}) \cdot \psi(X_1).$$

Here we denote by $\boldsymbol{Y} \in \mathbb{R}^{k \times \mathcal{Y}}, \boldsymbol{Y}_{:,y} = y$ that spans the whole support $\mathcal{Y}$. Therefore let $\boldsymbol{W}^* = \boldsymbol{Y}\boldsymbol{B}$ will finish the proof.

$\square$

*Proof of Theorem 3.6.* With Lemma 3.5 we know $e_{\text{apx}} = 0$, and therefore $\boldsymbol{W}^*\psi(X_1) \equiv f^*(X_1)$. Next from basic inequality and the same proof as in Theorem 3.3 we have:

$$\|\psi(\boldsymbol{X}_1)\boldsymbol{W}^* - \psi(\boldsymbol{X}_1)\hat{\boldsymbol{W}}\| \leq 2\|P_{\psi(\boldsymbol{X}_1)}\boldsymbol{N}\|_F$$

Notice $\mathcal{N}$ is a random noise matrix whose row vectors are independent samples from some centered distribution. Also we assumed $\mathbb{E}[\|N\|^2|\boldsymbol{X}_1] \leq \sigma^2$, i.e. $\mathbb{E}[\|N\|^2|N] \leq \sigma^2$. Also, $P_{\psi(\boldsymbol{X}_1)}$ is a projection to dimension $c$. From Lemma A.7 we have:

$$\|f^*(\boldsymbol{X}_1) - \psi(\boldsymbol{X}_1)\hat{\boldsymbol{W}}\| \leq \sigma\sqrt{c + \log c/\delta}.$$

Next, with Claim A.3 we have when $n \gg \rho^4(c + \log(1/\delta))$, since $\boldsymbol{W}^* - \hat{\boldsymbol{W}} \in \mathbb{R}^{d_2 \times k}$,

$$\begin{aligned} &0.9(\boldsymbol{W}^* - \hat{\boldsymbol{W}})^\top \boldsymbol{\Sigma}_\psi (\boldsymbol{W}^* - \hat{\boldsymbol{W}}) \\ &\preceq \frac{1}{n_2}(\boldsymbol{W}^* - \hat{\boldsymbol{W}})^\top \sum_i \psi(x_1^{(i)})\psi(x_1^{(i)})^\top (\boldsymbol{W}^* - \hat{\boldsymbol{W}}) \preceq 1.1(\boldsymbol{W}^* - \hat{\boldsymbol{W}})^\top \boldsymbol{\Sigma}_\psi (\boldsymbol{W}^* - \hat{\boldsymbol{W}}) \end{aligned}$$

And therefore we could easily conclude that:

$$\mathbb{E}\|\hat{\boldsymbol{W}}^\top \psi(X_1) - f^*(X_1)\|^2 \lesssim \sigma^2 \frac{c + \log(c/\delta)}{n_2}.$$

$\square$

## B.2 Omitted proof of linear model with approximation error

*Proof of Theorem 3.9.* First we note that $Y = f^*(X_1) + N$, where $\mathbb{E}[N|X_1] = 0$ but $Y - (\boldsymbol{A}^*)^\top X_1$ is not necessarily mean zero, and this is where additional difficulty lies. Write approximation error term $a(X_1) := f^*(X_1) - (\boldsymbol{A}^*)^\top X_1$, namely $Y = a(X_1) + (\boldsymbol{A}^*)^\top X_1 + N$. Also, $(\boldsymbol{A}^*)^\top X_1 \equiv (\boldsymbol{W}^*)^\top \psi(X_1)$ with conditional independence.

Second, with KKT condition on the training data, we know that $\mathbb{E}[a(X_1)X_1^\top] = 0$.

Recall $\hat{\boldsymbol{W}} = \arg\min_{\boldsymbol{W}} \|\boldsymbol{Y} - \psi(\boldsymbol{X}_1)\boldsymbol{W}\|_F^2$. We have the basic inequality,

$$\begin{aligned} \frac{1}{2n_2}\|\boldsymbol{Y} - \psi(\boldsymbol{X}_1)\hat{\boldsymbol{W}}\|_F^2 &\leq \frac{1}{2n_2}\|\boldsymbol{Y} - \boldsymbol{X}_1\boldsymbol{A}^*\|_F^2 \\ &= \frac{1}{2n_2}\|\boldsymbol{Y} - \psi(\boldsymbol{X}_1)\boldsymbol{W}^*\|_F^2. \end{aligned}$$

i.e., $\frac{1}{2n_2}\|\psi(\boldsymbol{X}_1)\boldsymbol{W}^* + a(\boldsymbol{X}_1) + \boldsymbol{N} - \psi(\boldsymbol{X}_1)\hat{\boldsymbol{W}}\|_F^2 \leq \frac{1}{2n_2}\|a(\boldsymbol{X}_1) + \boldsymbol{N}\|_F^2.$

Therefore

$$\begin{aligned} &\frac{1}{2n_2}\|\psi(\boldsymbol{X}_1)\boldsymbol{W}^* - \psi(\boldsymbol{X}_1)\hat{\boldsymbol{W}}\|^2 \\ &\leq -\frac{1}{n_2}\langle a(\boldsymbol{X}_1) + \boldsymbol{N}, \psi(\boldsymbol{X}_1)\boldsymbol{W}^* - \psi(\boldsymbol{X}_1)\hat{\boldsymbol{W}}\rangle \\ &= -\frac{1}{n_2}\langle a(\boldsymbol{X}_1), \psi(\boldsymbol{X}_1)\boldsymbol{W}^* - \psi(\boldsymbol{X}_1)\hat{\boldsymbol{W}}\rangle - \langle \boldsymbol{N}, \psi(\boldsymbol{X}_1)\boldsymbol{W}^* - \psi(\boldsymbol{X}_1)\hat{\boldsymbol{W}}\rangle \end{aligned} \quad (9)$$

With Assumption 3.6 and by concentration $0.9\frac{1}{n_2}\boldsymbol{X}_1\boldsymbol{X}_1^\top \preceq \boldsymbol{\Sigma}_{X_1} \preceq 1.1\frac{1}{n_2}\boldsymbol{X}_1\boldsymbol{X}_1^\top$, we have

$$\frac{1}{\sqrt{n_2}}\|a(\boldsymbol{X}_1)\boldsymbol{X}_1^\top \boldsymbol{\Sigma}_{X_1}^{-1/2}\|_F \leq 1.1b_0\sqrt{k} \quad (10)$$

Denote $\psi(\boldsymbol{X}_1) = \boldsymbol{X}_1\boldsymbol{B}$, where $\boldsymbol{B} = \boldsymbol{\Sigma}_{X_1}^{-1}\boldsymbol{\Sigma}_{X_1 X_2}$ is rank $k$ under exact CI since $\boldsymbol{\Sigma}_{X_1 X_2} = \boldsymbol{\Sigma}_{X_1 Y}\boldsymbol{\Sigma}_Y^{-1}\boldsymbol{\Sigma}_{Y X_2}$. We have

$$\frac{1}{n_2}\langle a(\boldsymbol{X}_1), \psi(\boldsymbol{X}_1)\boldsymbol{W}^* - \psi(\boldsymbol{X}_1)\hat{\boldsymbol{W}}\rangle$$

$$= \frac{1}{n_2}\langle a(\boldsymbol{X}_1), \boldsymbol{X}_1\boldsymbol{B}\boldsymbol{W}^* - \boldsymbol{X}_1\boldsymbol{B}\hat{\boldsymbol{W}}\rangle$$

$$= \frac{1}{n_2}\langle \boldsymbol{\Sigma}_{X_1}^{-1/2}\boldsymbol{X}_1^\top a(\boldsymbol{X}_1), \boldsymbol{\Sigma}_{X_1}^{1/2}(\boldsymbol{B}\boldsymbol{W}^* - \boldsymbol{B}\hat{\boldsymbol{W}})\rangle$$

$$\leq \sqrt{\frac{k}{n_2}}\|\boldsymbol{\Sigma}_{X_1}^{1/2}(\boldsymbol{B}\boldsymbol{W}^* - \boldsymbol{B}\hat{\boldsymbol{W}})\|_F \qquad \text{(from Ineq. (10))}$$

Back to Eqn. (9), we get

$$\frac{1}{2n_2}\|\psi(\boldsymbol{X}_1)\boldsymbol{W}^* - \psi(\boldsymbol{X}_1)\hat{\boldsymbol{W}}\|_F^2$$

$$\lesssim \sqrt{\frac{k}{n_2}}\|\boldsymbol{\Sigma}_{X_1}^{1/2}(\boldsymbol{B}\boldsymbol{W}^* - \boldsymbol{B}\hat{\boldsymbol{W}})\|_F + \frac{1}{n_2}\|P_{\boldsymbol{X}_1}\boldsymbol{N}\|_F\|\boldsymbol{X}_1(\boldsymbol{B}\boldsymbol{W}^* - \boldsymbol{B}\hat{\boldsymbol{W}})\|_F$$

$$\lesssim \left(\frac{\sqrt{k}}{n_2} + \frac{1}{n_2}\|P_{\boldsymbol{X}_1}\boldsymbol{N}\|_F\right)\|\boldsymbol{X}_1(\boldsymbol{B}\boldsymbol{W}^* - \boldsymbol{B}\hat{\boldsymbol{W}})\|_F$$

$$\Longrightarrow \frac{1}{\sqrt{n_2}}\|\psi(\boldsymbol{X}_1)\boldsymbol{W}^* - \psi(\boldsymbol{X}_1)\hat{\boldsymbol{W}}\|_F \lesssim \sqrt{\frac{k + \log k/\delta}{n_2}}.$$

Finally, by concentration we transfer the result from empirical loss to excess risk and get:

$$\mathbb{E}[\|\psi(X_1)\boldsymbol{W}^* - \psi(X_1)\hat{\boldsymbol{W}}\|^2] \lesssim \frac{k + \log(k/\delta)}{n_2}.$$

$\square$

### B.3 Argument on Denoising Auto-encoder or Context Encoder

**Remark B.1.** *We note that since $X_1 \perp X_2 | Y$ ensures $X_1 \perp h(X_2) | Y$ for any deterministic function $h$, we could replace $X_2$ by $h(X_2)$ and all results hold. Therefore in practice, we could use $h(\psi(X_1))$ instead of $\psi(X_1)$ for downstream task. Specifically with denoising auto-encoder or context encoder, one could think about $h$ as the inverse of decoder $D$ ($h = D^{-1}$) and use $D^{-1}\psi \equiv E$ the encoder function as the representation for downstream tasks, which is more commonly used in practice.*

This section explains what we claim in Remark B.1. For context encoder, the reconstruction loss targets to find the encoder $E^*$ and decoder $D^*$ that achieve

$$\min_E \min_D \mathbb{E}\|X_2 - D(E(X_1))\|_F^2, \qquad (11)$$

where $X_2$ is the masked part we want to recover and $X_1$ is the remainder.

If we naively apply our theorem we should use $D^*(E^*(\cdot))$ as the representation, while in practice we instead use only the encoder part $E^*(\cdot)$ as the learned representation. We argue that our theory also support this practical usage if we view the problem differently. Consider the pretext task to predict $(D^*)^{-1}(X_2)$ instead of $X_2$ directly, namely,

$$\bar{E} \leftarrow \arg\min_E \mathbb{E}\|(D^*)^{-1}(X_2) - E(X_1)\|^2, \qquad (12)$$

and then we should indeed use $E(X_1)$ as the representation. On one hand, when $X_1 \perp X_2 | Y$, it also satisfies $X_1 \perp (D^*)^{-1}(X_2) | Y$ since $(D^*)^{-1}$ is a deterministic function of $X_2$ and all our theory applies. On the other hand, the optimization on (11) or (12) give us similar result. Let

$$E^* = \arg\min_E \mathbb{E}[\|X_2 - D^*(E(X_1))\|^2],$$

and $\mathbb{E}\|X_2 - D^*(E^*(X_1))\|^2 \le \epsilon$, then with pretext task as in (12) we have that:

$$\begin{aligned}
\mathbb{E}\|(D^*)^{-1}(X_2) - E^*(X_1)\|^2 &= \mathbb{E}\|(D^*)^{-1}(X_2) - (D^*)^{-1} \circ D^*(E^*(X_1))\|^2 \\
&\le \|(D^*)^{-1}\|_{\mathrm{Lip}}^2 \, \mathbb{E}\|X_2 - D^*(E^*(X_1))\|^2 \\
&\le L^2 \epsilon,
\end{aligned}$$

where $L := \|(D^*)^{-1}\|_{\mathrm{Lip}}$ is the Lipschitz constant for function $(D^*)^{-1}$. This is to say, in practice, we optimize over (11), and achieves a good representation $E^*(X_1)$ such that $\epsilon_{\mathrm{pre}} \le L\sqrt{\epsilon}$ and thus performs well for downstream tasks. (Recall $\epsilon_{\mathrm{pre}}$ is defined in Theorem 4.2 that measures how well we have learned the pretext task.)

## C   OMITTED PROOFS BEYOND CONDITIONAL INDEPENDENCE

### C.1   WARM-UP: JOINTLY GAUSSIAN VARIABLES

As before, for simplicity we assume all data is centered in this case.

**Assumption C.1** (Approximate Conditional Independent Given Latent Variables). *Assume there exists some latent variable $Z \in \mathbb{R}^m$ such that*

$$\|\Sigma_{X_1}^{-1/2}\Sigma_{X_1,X_2|\bar{Y}}\|_F \le \epsilon_{CI},$$

$\sigma_{k+m}(\Sigma_{\bar{Y}\bar{Y}}^\dagger \Sigma_{\bar{Y}X_2}) = \beta > 0$ [3] *and* $\Sigma_{X_2,\bar{Y}}$ *is of rank* $k+m$, *where* $\bar{Y} = [Y, Z]$.

When $X_1$ is not exactly CI of $X_2$ given $Y$ and $Z$, the approximation error depends on the norm of $\|\Sigma_{X_1}^{-1/2}\Sigma_{X_1,X_2|\bar{Y}}\|_2$. Let $\hat{W}$ be the solution from Equation 2.

**Theorem C.1.** *Under Assumption C.1 with constant $\epsilon_{CI}$ and $\beta$, then the excess risk satisfies*

$$\mathrm{ER}_{\psi^*}[\hat{W}] := \mathbb{E}[\|\hat{W}^\top \psi^*(X_1) - f^*(X_1)\|_F^2] \lesssim \frac{\epsilon_{CI}^2}{\beta^2} + \mathrm{Tr}(\Sigma_{YY|X_1})\frac{d_2 + \log(d_2/\delta)}{n_2}.$$

*Proof of Theorem C.1.* Let $\boldsymbol{V} := f^*(\boldsymbol{X}_1) \equiv \boldsymbol{X}_1\Sigma_{X_1X_1}^{-1}\Sigma_{1Y}$ be our target direction. Denote the optimal representation matrix by $\Psi := \psi(\boldsymbol{X}_1) \equiv \boldsymbol{X}_1\boldsymbol{A}$ (where $\boldsymbol{A} := \Sigma_{X_1X_1}^{-1}\Sigma_{X_1X_2}$).

Next we will make use of the conditional covariance matrix:

$$\Sigma_{X_1X_2|\bar{Y}} := \Sigma_{X_1X_2} - \Sigma_{X_1\bar{Y}}\Sigma_{\bar{Y}}^{-1}\Sigma_{\bar{Y}X_2},$$

and plug it in into the definition of $\Psi$:

$$\begin{aligned}
\Psi &= \boldsymbol{X}_1\Sigma_{X_1X_1}^{-1}\Sigma_{X_1\bar{Y}}\Sigma_{\bar{Y}}^{-1}\Sigma_{\bar{Y}X_2} + \boldsymbol{X}_1\Sigma_{X_1X_1}^{-1}\Sigma_{X_1X_2|\bar{Y}} \\
&=: \boldsymbol{L} + \boldsymbol{E},
\end{aligned}$$

where $\boldsymbol{L} := \boldsymbol{X}_1\Sigma_{X_1X_1}^{-1}\Sigma_{X_1\bar{Y}}\Sigma_{\bar{Y}}^{-1}\Sigma_{\bar{Y}X_2}$ and $\boldsymbol{E} := \boldsymbol{X}_1\Sigma_{X_1X_1}^{-1}\Sigma_{X_1X_2|\bar{Y}}$. We analyze these two terms respectively.

For $\boldsymbol{L}$, we note that $\mathrm{span}(\boldsymbol{V}) \subseteq \mathrm{span}(\boldsymbol{L})$: $\boldsymbol{L}\Sigma_{X_2\bar{Y}}^\dagger\Sigma_{\bar{Y}} = \boldsymbol{X}_1\Sigma_{X_1X_1}^{-1}\Sigma_{X_1\bar{Y}}$. By right multiplying the selector matrix $S_Y$ we have: $\boldsymbol{L}\Sigma_{X_2\bar{Y}}^\dagger\Sigma_{\bar{Y}Y} = \boldsymbol{X}_1\Sigma_{X_1X_1}^{-1}\Sigma_{X_1Y}$, i.e., $\boldsymbol{L}\bar{W} = \boldsymbol{V}$, where $\bar{W} := \Sigma_{X_2\bar{Y}}^\dagger\Sigma_{\bar{Y}Y}$. From our assumption that $\sigma_r(\Sigma_{\bar{Y}Y}^\dagger\Sigma_{\bar{Y}X_2}) = \beta$, we have $\|\bar{W}\|_2 \le \|\Sigma_{X_2\bar{Y}}^\dagger\Sigma_{\bar{Y}}\|_2 \le 1/\beta$. (Or we could directly define $\beta$ as $\sigma_k(\Sigma_{Y\bar{Y}}^\dagger\Sigma_{\bar{Y}X_2}) \equiv \|\bar{W}\|_2$.)

By concentration, we have $\boldsymbol{E} = \boldsymbol{X}_1\Sigma_{X_1X_1}^{-1}\Sigma_{X_1X_2|\bar{Y}}$ converges to $\Sigma_{X_1X_1}^{-1/2}\Sigma_{X_1X_2|\bar{Y}}$. Specifically, when $n \gg k + \log 1/\delta$, $\|\boldsymbol{E}\|_F \le 1.1\|\Sigma_{X_1X_1}^{-1/2}\Sigma_{X_1X_2|\bar{Y}}\|_F \le 1.1\epsilon_{CI}$ (by using Lemma A.2 ). Together we have $\|\boldsymbol{E}\bar{W}\|_F \lesssim \epsilon_{CI}/\beta$.

Let $\hat{W} = \arg\min_{\boldsymbol{W}} \|\boldsymbol{Y} - \Psi\boldsymbol{W}\|^2$. We note that $\boldsymbol{Y} = \boldsymbol{N} + \boldsymbol{V} = \boldsymbol{N} + \Psi\bar{W} - \boldsymbol{E}\bar{W}$ where $\boldsymbol{V}$ is our target direction and $\boldsymbol{N}$ is random noise (each row of $\boldsymbol{N}$ has covariance matrix $\Sigma_{YY|X_1}$).

---

[3] $\sigma_k(\boldsymbol{A})$ denotes $k$-th singular value of $\boldsymbol{A}$, and $\boldsymbol{A}^\dagger$ is the pseudo-inverse of $\boldsymbol{A}$.

From basic inequality, we have:

$$\|\Psi\hat{\boldsymbol{W}} - \boldsymbol{Y}\|_F^2 \le \|\Psi\bar{\boldsymbol{W}} - \boldsymbol{Y}\|_F^2 = \|\boldsymbol{N} - \boldsymbol{E}\bar{\boldsymbol{W}}\|_F^2.$$

$$\implies \|\Psi\hat{\boldsymbol{W}} - \boldsymbol{V} - \boldsymbol{E}\bar{\boldsymbol{W}}\|^2 \le 2\langle\Psi\hat{\boldsymbol{W}} - \boldsymbol{V} - \boldsymbol{E}\bar{\boldsymbol{W}}, \boldsymbol{N} - \boldsymbol{E}\bar{\boldsymbol{W}}\rangle$$

$$\implies \|\Psi\hat{\boldsymbol{W}} - \boldsymbol{V} - \boldsymbol{E}\bar{\boldsymbol{W}}\| \le \|P_{[\Psi,E,V]}\boldsymbol{N}\| + \|\boldsymbol{E}\bar{\boldsymbol{W}}\|$$

$$\implies \|\Psi\hat{\boldsymbol{W}} - \boldsymbol{V}\| \lesssim \|\boldsymbol{E}\|_F\|\bar{\boldsymbol{W}}\| + (\sqrt{d_2} + \sqrt{\log 1/\delta})\sqrt{\text{Tr}(\boldsymbol{\Sigma}_{YY|X_1})}.$$
$$\text{(from Lemma A.7)}$$

$$\le \sqrt{n_2}\frac{\epsilon_{\text{CI}}}{\beta} + (\sqrt{d_2} + \sqrt{\log 1/\delta})\sqrt{\text{Tr}(\boldsymbol{\Sigma}_{YY|X_1})}.$$
$$\text{(from Assumption C.1)}$$

Next, by the same procedure that concentrates $\frac{1}{n_2}\boldsymbol{X}_1^\top\boldsymbol{X}_1$ to $\boldsymbol{\Sigma}_{X_1X_1}$ with Claim A.2, we could easily get

$$\text{ER}[\hat{\boldsymbol{W}}] := \mathbb{E}[\|\hat{\boldsymbol{W}}^\top\psi(X_1) - f^*(X_1)\|^2] \lesssim \frac{\epsilon_{\text{CI}}^2}{\beta^2} + \text{Tr}(\boldsymbol{\Sigma}_{YY|X_1})\frac{d_2 + \log 1/\delta}{n_2}.$$

$$\square$$

## C.2 TECHNICAL FACTS

**Lemma C.2** (Approximation Error of PCA). *Let matrix $\boldsymbol{A} = \boldsymbol{L} + \boldsymbol{E}$ where $\boldsymbol{L}$ is rank $r <$ size of $\boldsymbol{A}$ and $\|\boldsymbol{E}\|_2 \le \epsilon$ and $\boldsymbol{\Sigma}_r(\boldsymbol{A}) = \beta$. Then we have*

$$\|\sin\Theta(\boldsymbol{A},\boldsymbol{L})\|_2 \le \epsilon/\beta.$$

*Proof.* We use Davis Kahan for this proof. First note that $\|\boldsymbol{A} - \boldsymbol{L}\| = \|\boldsymbol{E}\| \le \epsilon$. From Davis-Kahan we get:

$$\|\sin\Theta(\boldsymbol{A},\boldsymbol{L})\|_2 \le \frac{\|\boldsymbol{E}\|_2}{\boldsymbol{\Sigma}_r(\boldsymbol{A}) - \boldsymbol{\Sigma}_{r+1}(\boldsymbol{L})}$$
$$= \frac{\|\boldsymbol{E}\|_2}{\boldsymbol{\Sigma}_r(\boldsymbol{A})}$$
$$\lesssim \epsilon/\beta.$$

$$\square$$

## C.3 MEASURING CONDITIONAL DEPENDENCE WITH CROSS-COVARIANCE OPERATOR

$L^2(P_X)$ denotes the Hilbert space of square integrable function with respect to the measure $P_X$, the marginal distribution of $X$. We are interested in some function class $\mathcal{H}_x \subset L^2(P_X)$ that is induced from some feature maps:

**Definition C.3** (General and Universal feature Map). *We denote feature map $\phi : \mathcal{X} \to \mathcal{F}$ that maps from a compact input space $\mathcal{X}$ to the feature space $\mathcal{F}$. $\mathcal{F}$ is a Hilbert space associated with inner product: $\langle\phi(\boldsymbol{x}), \phi(\boldsymbol{x}')\rangle_\mathcal{F}$. The associated function class is: $\mathcal{H}_x = \{h : \mathcal{X} \to \mathbb{R}|\exists w \in \mathcal{F}, h(\boldsymbol{x}) = \langle w, \phi(\boldsymbol{x})\rangle_\mathcal{F}, \forall \boldsymbol{x} \in \mathcal{X}\}$. We call $\phi$ universal if the induced $\mathcal{H}_x$ is dense in $L^2(P_X)$.*

Linear model is a special case when feature map $\phi = Id$ is identity mapping and the inner product is over Euclidean space. A feature map with higher order polynomials correspondingly incorporate high order moments (Fukumizu et al., 2004; Gretton et al., 2005). For discrete variable $Y$ we overload $\phi$ as the one-hot embedding.

**Remark C.1.** *For continuous data, any universal kernel like Gaussian kernel or RBF kernel induce the universal feature map that we require (Micchelli et al., 2006). Two-layer neural network with infinite width also satisfy it, i.e., $\forall\boldsymbol{x} \in \mathcal{X} \subset \mathbb{R}^d, \phi_{NN}(\boldsymbol{x}) : \mathcal{S}^{d-1} \times \mathbb{R} \to \mathbb{R}, \phi_{NN}(\boldsymbol{x})[\boldsymbol{w}, b] = \sigma(\boldsymbol{w}^\top\boldsymbol{x} + b)$ (Barron, 1993).*

When there's no ambiguity, we overload $\phi_1$ as the random variable $\phi_1(X_1)$ over domain $\mathcal{F}_1$, and $\mathcal{H}_1$ as the function class over $X_1$. Next we characterize CI using the cross-covariance operator.

**Definition C.4** (Cross-covariance operator). *For random variables $X \in \mathcal{X}, Y \in \mathcal{Y}$ with joint distribution $P : \mathcal{X} \times \mathcal{Y} \to \mathbb{R}$, and associated feature maps $\phi_x$ and $\phi_y$, we denote by $\mathcal{C}_{\phi_x \phi_y} = \mathbb{E}[\phi_x(X) \otimes \phi_y(Y)] = \int_{\mathcal{X} \times \mathcal{Y}} \phi_x(x) \otimes \phi_y(y) dP(x, y)$, the (un-centered) cross-covariance operator. Similarly we denote by $\mathcal{C}_{X\phi_y} = \mathbb{E}[X \otimes \phi_y(Y)] : \mathcal{F}_y \to \mathcal{X}$.*

To understand what $\mathcal{C}_{\phi_x \phi_y}$ is, we note it is of the same shape as $\phi_x(x) \otimes \phi_y(y)$ for each individual $x \in \mathcal{X}, y \in \mathcal{Y}$. It can be viewed as a self-adjoint operator: $\mathcal{C}_{\phi_x \phi_y} : \mathcal{F}_y \to \mathcal{F}_x$, $\mathcal{C}_{\phi_x \phi_y} f = \int_{\mathcal{X} \times \mathcal{Y}} \langle \phi_y(y), f \rangle \phi_x(x) dP(x, y), \forall f \in \mathcal{F}_y$. For any $f \in \mathcal{H}_x$ and $g \in \mathcal{H}_y$, it satisfies: $\langle f, \mathcal{C}_{\phi_x \phi_y} g \rangle_{\mathcal{H}_x} = \mathbb{E}_{XY}[f(X)g(Y)]$(Baker, 1973; Fukumizu et al., 2004). CI ensures $\mathcal{C}_{\phi_1 X_2 | \phi_y} = 0$ for arbitrary $\phi_1, \phi_2$:

**Lemma C.5.** *With one-hot encoding map $\phi_y$ and arbitrary $\phi_1$, $X_1 \perp X_2 | Y$ ensures:*

$$\mathcal{C}_{\phi_1 X_2 | \phi_y} := \mathcal{C}_{\phi_1 X_2} - \mathcal{C}_{\phi_1 \phi_y} \mathcal{C}_{\phi_y \phi_y}^{-1} \mathcal{C}_{\phi_y X_2} = 0. \tag{13}$$

A more complete discussion of cross-covariance operator and CI can be found in (Fukumizu et al., 2004). Also, recall that an operator $\mathcal{C} : \mathcal{F}_y \to \mathcal{F}_x$ is Hilbert-Schmidt (HS) (Reed, 2012) if for complete orthonormal systems (CONSs) $\{\zeta_i\}$ of $\mathcal{F}_x$ and $\{\eta_i\}$ of $\mathcal{F}_y$, $\|\mathcal{C}\|_{\text{HS}}^2 := \sum_{i,j} \langle \zeta_j, \mathcal{C} \eta_i \rangle_{\mathcal{F}_x}^2 < \infty$. The Hilbert-Schmidt norm generalizes the Frobenius norm from matrices to operators, and we will later use $\|\mathcal{C}_{\phi_1 X_2 | \phi_y}\|$ to quantify approximate CI.

We note that covariance operators (Fukumizu et al., 2009; 2004; Baker, 1973) are commonly used to capture conditional dependence of random variables. In this work, we utilize the covariance operator to quantify the performance of the algorithm even when the algorithm is *not a kernel method*.

## C.4 Omitted Proof in General Setting

**Claim C.6.** *For feature maps $\phi_1$ with universal property, we have:*

$$\psi^*(X_1) := \mathbb{E}[X_2 | X_1] = \mathbb{E}^L[X_2 | \phi_1]$$
$$= \mathcal{C}_{X_2 \phi_1} \mathcal{C}_{\phi_1 \phi_1}^{-1} \phi_1(X_1).$$
$$\textit{Our target } f^*(X_1) := \mathbb{E}[Y | X_1] = \mathbb{E}^L[Y | \phi_1]$$
$$= \mathcal{C}_{Y \phi_1} \mathcal{C}_{\phi_1 \phi_1}^{-1} \phi_1(X_1).$$

*For general feature maps, we instead have:*

$$\psi^*(X_1) := \arg\min_{f \in \mathcal{H}_1^{d_2}} \mathbb{E}_{X_1 X_2} \|X_2 - f(X_1)\|_2^2$$
$$= \mathcal{C}_{X_2 \phi_1} \mathcal{C}_{\phi_1 \phi_1}^{-1} \phi_1(X_1).$$
$$\textit{Our target } f^*(X_1) := \arg\min_{f \in \mathcal{H}_1^k} \mathbb{E}_{X_1 Y} \|Y - f(X_1)\|_2^2$$
$$= \mathcal{C}_{Y \phi_1} \mathcal{C}_{\phi_1 \phi_1}^{-1} \phi_1(X_1).$$

To prove Claim C.6, we show the following lemma:

**Lemma C.7.** *Let $\phi : \mathcal{X} \to \mathcal{F}_x$ be a universal feature map, then for random variable $Y \in \mathcal{Y}$ we have:*

$$\mathbb{E}[Y | X] = \mathbb{E}^L[Y | \phi(X)].$$

*Proof of Lemma C.7.* Denote by $\mathbb{E}[Y | X = x] =: f(x)$. Since $\phi$ is dense in $\mathcal{X}$, there exists a linear operator $a : \mathcal{X} \to \mathbb{R}$ such that $\int_{x \in \mathcal{X}} a(x) \phi(x)[\cdot] dx = f(\cdot)$ a.e. Therefore the result comes directly from the universal property of $\phi$. □

*Proof of Claim C.6.* We want to show that for random variables $Y, X$, where $X$ is associated with a universal feature map $\phi_x$, we have $\mathbb{E}[Y | X] = \mathcal{C}_{Y \phi_x(X)} \mathcal{C}_{\phi_x(X) \phi_x(X)}^{-1} \phi_x(X)$.

First, from Lemma C.7, we have that $\mathbb{E}[Y|X] = \mathbb{E}^L[Y|\phi_x(X)]$. Next, write $A^* : \mathcal{F}_x \to \mathcal{Y}$ as the linear operator that satisfies

$$\mathbb{E}[Y|X] = A^*\phi_x(X)$$
$$\text{s.t. } A^* = \arg\min_A \mathbb{E}[\|Y - A\phi_x(X)\|^2].$$

Therefore from the stationary condition we have $A^*\mathbb{E}_X[\phi_x(X) \otimes \phi_x(X)] = \mathbb{E}_{XY}[Y \otimes \phi_x(X)]$. Or namely we get $A^* = \mathcal{C}_{Y\phi_x}\mathcal{C}_{\phi_x\phi_x}^{-1}$ simply from the definition of the cross-covariance operator $\mathcal{C}$. $\qquad\square$

**Claim C.8.** $\|\mathcal{C}_{\phi_1\phi_1}^{-1/2}\mathcal{C}_{\phi_1 X_2|\phi_{\bar{y}}}\|_{HS}^2 = \mathbb{E}_{X_1}[\|\mathbb{E}[X_2|X_1] - \mathbb{E}_{\bar{Y}}[\mathbb{E}[X_2|\bar{Y}]|X_1]\|^2] = \epsilon_{CI}^2.$

*Proof.*

$$\|\mathcal{C}_{\phi_1\phi_1}^{-1/2}\mathcal{C}_{\phi_1 X_2|\phi_{\bar{y}}}\|_{HS}^2$$
$$= \int_{X_1} \left\|\int_{X_2}\left(\frac{p_{X_1 X_2}(\boldsymbol{x}_1, \boldsymbol{x}_2)}{p_{X_1}(\boldsymbol{x}_1)} - \frac{p_{X_1 \perp X_2|Y}(\boldsymbol{x}_1, \boldsymbol{x}_2)}{p_{X_1}(\boldsymbol{x}_1)}\right)X_2 dp_{\boldsymbol{x}_2}\right\|^2 dp_{\boldsymbol{x}_1}$$
$$= \mathbb{E}_{X_1}[\|\mathbb{E}[X_2|X_1] - \mathbb{E}_{\bar{Y}}[\mathbb{E}[X_2|\bar{Y}]|X_1]\|^2].$$

$\qquad\square$

## C.5 OMITTED PROOF FOR MAIN RESULTS

We first prove a simpler version without approximation error.

**Theorem C.9.** *For a fixed $\delta \in (0, 1)$, under Assumption 4.1, 3.5, if there is no approximation error, i.e., there exists a linear operator $A$ such that $f^*(X_1) \equiv A\phi_1(X_1)$, if $n_1, n_2 \gg \rho^4(d_2 + \log 1/\delta)$, and we learn the pretext tasks such that:*

$$\mathbb{E}\|\tilde{\psi}(X_1) - \psi^*(X_1)\|_F^2 \le \epsilon_{pre}^2.$$

*Then we are able to achieve generalization for downstream task with probability $1 - \delta$:*

$$\mathbb{E}[\|f_{\mathcal{H}_1}^*(X_1) - \hat{\boldsymbol{W}}^\top\tilde{\psi}(X_1)\|^2] \le \mathcal{O}\{\sigma^2\frac{d_2 + \log d_2/\delta}{n_2} + \frac{\epsilon_{CI}^2}{\beta^2} + \frac{\epsilon_{pre}^2}{\beta^2}\}. \qquad (14)$$

*Proof of Theorem C.9.* We follow the similar procedure as Theorem C.1. For the setting of no approximation error, we have $f^* = f_{\mathcal{H}_1}^*$, and the residual term $N := Y - f^*(X_1)$ is a mean-zero random variable with $\mathbb{E}[\|N\|^2|X_1] \lesssim \sigma^2$ according to our data assumption in Section 3. $\boldsymbol{N} = \boldsymbol{Y} - f^*(\boldsymbol{X}_1^{\text{down}})$ is the collected $n_2$ samples of noise terms. We write $Y \in \mathbb{R}^{d_3}$. For classification task, we have $Y \in \{\boldsymbol{e}_i, i \in [k]\} \subset \mathbb{R}^k$ (i.e, $d_3 = k$) is one-hot encoded random variable. For regression problem, $Y$ might be otherwise encoded. For instance, in the yearbook dataset, Y ranges from 1905 to 2013 and represents the years that the photos are taken. We want to note that our result is general for both cases: the bound doesn't depend on $d_3$, but only depends on the variance of $N$.

Let $\Psi^*, \boldsymbol{L}, \boldsymbol{E}, \boldsymbol{V}$ be defined as follows:

Let $\boldsymbol{V} = f^*(\boldsymbol{X}_1^{\text{down}}) \equiv f_{\mathcal{H}_1}^*(\boldsymbol{X}_1^{\text{down}}) \equiv \phi(\boldsymbol{X}_1^{\text{down}})\mathcal{C}_{\phi_1}^{-1}\mathcal{C}_{\phi_1 Y}$ be our target direction. Denote the optimal representation matrix by

$$\Psi^* := \psi^*(\boldsymbol{X}_1^{\text{down}})$$
$$= \phi(\boldsymbol{X}_1^{\text{down}})\mathcal{C}_{\phi_1\phi_1}^{-1}\mathcal{C}_{\phi_1 X_2}$$
$$= \phi(\boldsymbol{X}_1^{\text{down}})\mathcal{C}_{\phi_1\phi_1}^{-1}\mathcal{C}_{\phi_1\phi_{\bar{y}}}\mathcal{C}_{\phi_{\bar{y}}}^{-1}\boldsymbol{\Sigma}_{\phi_{\bar{y}}X_2} + \phi(\boldsymbol{X}_1^{\text{down}})\mathcal{C}_{\phi_1\phi_1}^{-1}\mathcal{C}_{\phi_1 X_2|\phi_{\bar{y}}}$$
$$=: \boldsymbol{L} + \boldsymbol{E},$$

where $\boldsymbol{L} = \phi(\boldsymbol{X}_1^{\text{down}})\mathcal{C}_{\phi_1\phi_1}^{-1}\mathcal{C}_{\phi_1\phi_{\bar{y}}}\mathcal{C}_{\phi_{\bar{y}}}^{-1}\mathcal{C}_{\phi_{\bar{y}}X_2}$ and $\boldsymbol{E} = \phi(\boldsymbol{X}_1^{\text{down}})\mathcal{C}_{\phi_1\phi_1}^{-1}\mathcal{C}_{\phi_1 X_2|\bar{Y}}$.

In this proof, we denote $S_Y$ as the matrix such that $S_Y\phi_{\bar{y}} = Y$. Specifically, if $Y$ is of dimension $d_3$, $S_Y$ is of size $d_3 \times |\mathcal{Y}||\mathcal{Z}|$. Therefore $S_Y\boldsymbol{\Sigma}_{\phi_y A} = \boldsymbol{\Sigma}_{YA}$ for any random variable $A$.

Therefore, similarly we have:

$$\boldsymbol{L}\boldsymbol{\Sigma}_{X_2\phi_{\bar{y}}}^{\dagger}\boldsymbol{\Sigma}_{\phi_{\bar{y}}\phi_{\bar{y}}}S_Y^{\top} = \boldsymbol{L}\boldsymbol{\Sigma}_{X_2\phi_{\bar{y}}}^{\dagger}\boldsymbol{\Sigma}_{\phi_{\bar{y}}Y} = \boldsymbol{L}\bar{\boldsymbol{W}} = \boldsymbol{V}$$

where $\bar{\boldsymbol{W}} := \boldsymbol{\Sigma}_{X_2\phi_{\bar{y}}}^{\dagger}\boldsymbol{\Sigma}_{\phi_{\bar{y}}Y}$ satisfies $\|\bar{\boldsymbol{W}}\|_2 = 1/\beta$. Therefore span$(\boldsymbol{V}) \subseteq$span$(\boldsymbol{L})$ since we have assumed that $\boldsymbol{\Sigma}_{X_2\phi_{\bar{y}}}^{\dagger}\boldsymbol{\Sigma}_{\phi_{\bar{y}}Y}$ to be full rank.

On the other hand, $\boldsymbol{E} = \boldsymbol{X}_1^{\text{down}}\mathcal{C}_{\phi_1\phi_1}^{-1}\mathcal{C}_{\phi_1 X_2|\bar{Y}}$ concentrates to $\mathcal{C}_{\phi_1\phi_1}^{-1/2}\mathcal{C}_{\phi_1 X_2|\phi_{\bar{y}}}$. Specifically, when $n \gg c + \log 1/\delta$, $\|\boldsymbol{E}\|_F \leq 1.1\|\mathcal{C}_{\phi_1\phi_1}^{-1/2}\mathcal{C}_{\phi_1 X_2|\phi_{\bar{y}}}\|_F \leq 1.1\epsilon_{\text{CI}}$ (by using Lemma A.3 ). Together we have $\|\boldsymbol{E}\bar{\boldsymbol{W}}\|_F \lesssim \epsilon_{\text{CI}}/\beta$.

We also introduce the error from not learning $\psi^*$ exactly: $\boldsymbol{E}^{\text{pre}} = \Psi - \Psi^* := \tilde{\psi}(\boldsymbol{X}_1^{\text{down}}) - \psi^*(\boldsymbol{X}_1^{\text{down}})$. With proper concentration and our assumption, we have that $\mathbb{E}\|\psi(X_1) - \psi^*(X_1)\|^2 \leq \epsilon_{\text{pre}}$ and $\frac{1}{\sqrt{n_2}}\psi(\boldsymbol{X}_1^{\text{down}}) - \psi^*(\boldsymbol{X}_1^{\text{down}})\|^2 \leq 1.1\epsilon_{\text{pre}}$.

Also, the noise term after projection satisfies $\|P_{[\Psi,\boldsymbol{E},\boldsymbol{V}]}\boldsymbol{N}\| \lesssim \sqrt{d_2 + \log d_2/\delta}\sigma$ as using Lemma A.7. Therefore $\Psi = \Psi^* - \boldsymbol{E}^{\text{pre}} = \boldsymbol{L} + \boldsymbol{E} - \boldsymbol{E}^{\text{pre}}$.

Recall that $\hat{\boldsymbol{W}} = \arg\min_{\boldsymbol{W}} \|\psi(\boldsymbol{X}_1^{\text{down}})\boldsymbol{W} - \boldsymbol{Y}\|_F^2$. And with exactly the same procedure as Theorem C.1 we also get that:

$$\begin{aligned}
\|\Psi\hat{\boldsymbol{W}} - \boldsymbol{V}\| &\leq 2\|\boldsymbol{E}\bar{\boldsymbol{W}}\| + 2\|\boldsymbol{E}^{\text{pre}}\bar{\boldsymbol{W}}\| + \|P_{[\Psi,\boldsymbol{E},\boldsymbol{V},\boldsymbol{E}^{\text{pre}}]}\boldsymbol{N}\| \\
&\lesssim \sqrt{n_2}\frac{\epsilon_{\text{CI}} + \epsilon_{\text{pre}}}{\beta} + \sigma\sqrt{d_2 + \log(d_2/\delta)}.
\end{aligned}$$

With the proper concentration we also get:

$$\mathbb{E}[\|\hat{\boldsymbol{W}}^{\top}\psi(X_1) - f_{\mathcal{H}_1}^*(X_1)\|^2] \lesssim \frac{\epsilon_{\text{CI}}^2 + \epsilon_{\text{pre}}^2}{\beta^2} + \sigma^2\frac{d_2 + \log(d_2/\delta)}{n_2}.$$

$\qquad\qquad\qquad\qquad\qquad\qquad\qquad\qquad\qquad\qquad\qquad\qquad\qquad\qquad\qquad\qquad\quad\square$

Next we move on to the proof of our main result Theorem 4.2 where approximation error occurs.

*Proof of Theorem 4.2.* The proof is a combination of Theorem 3.9 and Theorem C.9. We follow the same notation as in Theorem C.9. Now the only difference is that an additional term $a(\boldsymbol{X}_1^{\text{down}})$ is included in $\boldsymbol{Y}$:

$$\begin{aligned}
\boldsymbol{Y} &= \boldsymbol{N} + f^*(\boldsymbol{X}_1^{\text{down}}) \\
&= \boldsymbol{N} + \Psi^*\bar{\boldsymbol{W}} + a(\boldsymbol{X}_1^{\text{down}}) \\
&= \boldsymbol{N} + (\Psi + \boldsymbol{E}^{\text{pre}})\bar{\boldsymbol{W}} + a(\boldsymbol{X}_1^{\text{down}}) \\
&= \Psi\bar{\boldsymbol{W}} + (\boldsymbol{N} + \boldsymbol{E}^{\text{pre}}\bar{\boldsymbol{W}} + a(\boldsymbol{X}_1^{\text{down}})).
\end{aligned}$$

From re-arranging $\frac{1}{2n_2}\|\boldsymbol{Y} - \Psi\hat{\boldsymbol{W}}\|_F^2 \leq \frac{1}{2n_2}\|\boldsymbol{Y} - \Psi\bar{\boldsymbol{W}}\|_F^2$,

$$\frac{1}{2n_2}\|\Psi(\bar{\boldsymbol{W}} - \hat{\boldsymbol{W}}) + (\boldsymbol{N} + \boldsymbol{E}^{\text{pre}} + a(\boldsymbol{X}_1^{\text{down}}))\|_F^2 \leq \frac{1}{2n_2}\|\boldsymbol{N} + \boldsymbol{E}^{\text{pre}}\bar{\boldsymbol{W}} + a(\boldsymbol{X}_1^{\text{down}})\|_F^2 \quad (15)$$

$$\Rightarrow \frac{1}{2n_2}\|\Psi(\bar{\boldsymbol{W}} - \hat{\boldsymbol{W}})\|_F^2 \leq \frac{1}{n_2}\langle\Psi(\bar{\boldsymbol{W}} - \hat{\boldsymbol{W}}), \boldsymbol{N} + \boldsymbol{E}^{\text{pre}}\bar{\boldsymbol{W}} + a(\boldsymbol{X}_1^{\text{down}})\rangle. \quad (16)$$

Then with similar procedure as in the proof of Theorem 3.9, and write $\Psi$ as $\phi(X_1^{\text{down}})\boldsymbol{B}$, we have:

$$\frac{1}{n_2}\langle\Psi(\bar{\boldsymbol{W}}-\hat{\boldsymbol{W}}),a(\boldsymbol{X}_1^{\text{down}})\rangle$$

$$=\frac{1}{n_2}\langle\boldsymbol{B}(\bar{\boldsymbol{W}}-\hat{\boldsymbol{W}}),\phi(\boldsymbol{X}_1^{\text{down}})^\top a(\boldsymbol{X}_1^{\text{down}})\rangle$$

$$=\frac{1}{n_2}\langle\mathcal{C}_{\phi_1}^{1/2}\boldsymbol{B}(\bar{\boldsymbol{W}}-\hat{\boldsymbol{W}}),\mathcal{C}_{\phi_1}^{-1/2}\phi(\boldsymbol{X}_1^{\text{down}})^\top a(\boldsymbol{X}_1^{\text{down}})\rangle$$

$$\leq\sqrt{\frac{d_2}{n_2}}\|\mathcal{C}_{\phi_1}^{1/2}\boldsymbol{B}(\bar{\boldsymbol{W}}-\hat{\boldsymbol{W}})\|_F$$

$$\leq 1.1\frac{1}{\sqrt{n_2}}\sqrt{\frac{d_2}{n_2}}\|\phi(\boldsymbol{X}_1^{\text{down}})\boldsymbol{B}(\bar{\boldsymbol{W}}-\hat{\boldsymbol{W}})\|_F$$

$$=1.1\frac{\sqrt{d_2}}{n_2}\|\Psi(\bar{\boldsymbol{W}}-\hat{\boldsymbol{W}})\|_F.$$

Therefore plugging back to (16) we get:

$$\frac{1}{2n_2}\|\Psi(\bar{\boldsymbol{W}}-\hat{\boldsymbol{W}})\|_F^2\leq\frac{1}{n_2}\langle\Psi(\bar{\boldsymbol{W}}-\hat{\boldsymbol{W}}),\boldsymbol{N}+\boldsymbol{E}^{\text{pre}}\bar{\boldsymbol{W}}+a(\boldsymbol{X}_1^{\text{down}})\rangle$$

$$\Rightarrow\frac{1}{2n_2}\|\Psi(\bar{\boldsymbol{W}}-\hat{\boldsymbol{W}})\|_F\leq\frac{1}{2n_2}\|\boldsymbol{E}^{\text{pre}}\bar{\boldsymbol{W}}\|_F+\frac{1}{2n_2}\|P_\Psi\boldsymbol{N}\|_F+1.1\frac{\sqrt{d_2}}{n_2}.$$

$$\Rightarrow\frac{1}{2\sqrt{n_2}}\|\Psi\hat{\boldsymbol{W}}-f_{\mathcal{H}_1}^*(\boldsymbol{X}_1^{\text{down}})\|_F-\|\boldsymbol{E}\bar{\boldsymbol{W}}\|_F\leq\frac{1}{\sqrt{n_2}}(1.1\sqrt{d_2}+\|\boldsymbol{E}^{\text{pre}}\bar{\boldsymbol{W}}\|+\sqrt{d_2+\log(d_2/\delta)})$$

$$\Rightarrow\frac{1}{2\sqrt{n_2}}\|\Psi\hat{\boldsymbol{W}}-f_{\mathcal{H}_1}^*(\boldsymbol{X}_1^{\text{down}})\|_F\lesssim\sqrt{\frac{d_2+\log d_2/\delta}{n_2}}+\frac{\epsilon_{\text{CI}}+\epsilon_{\text{pre}}}{\beta}.$$

Finally by concentrating $\frac{1}{n_2}\Psi^\top\Psi$ to $\mathbb{E}[\tilde{\psi}(X_1)\tilde{\psi}(X_1)^\top]$ we get:

$$\mathbb{E}[\|\hat{\boldsymbol{W}}^\top\tilde{\psi}(X_1)-f_{\mathcal{H}_1}^*(X_1)\|_2^2]\lesssim\frac{d_2+\log d_2/\delta}{n_2}+\frac{\epsilon_{\text{CI}}^2+\epsilon_{\text{pre}}^2}{\beta^2},$$

with probability $1-\delta$. $\qquad\square$

# D  THEORETICAL ANALYSIS FOR CLASSIFICATION TASKS

## D.1  CLASSIFICATION TASKS

We now consider the benefit of learning $\psi$ from a class $\mathcal{H}_1$ on linear classification task for label set $\mathcal{Y}=[k]$. The performance of a classifier is measured using the standard logistic loss

**Definition D.1.** *For a task with $\mathcal{Y}=[k]$, classification loss for a predictor $f:\mathcal{X}_1\to\mathbb{R}^k$ is*

$$\ell_{clf}(f)=\mathbb{E}[\ell_{log}(f(X_1),Y)],\text{ where }\ell_{log}(\hat{y},y)=\left[-\log\left(\frac{e^{\hat{y}_y}}{\sum_{y'}e^{\hat{y}_{y'}}}\right)\right]$$

*The loss for representation $\psi:\mathcal{X}_1\to\mathbb{R}^{d_1}$ and linear classifier $\boldsymbol{W}\in\mathbb{R}^{k\times d_1}$ is denoted by $\ell_{clf}(\boldsymbol{W}\psi)$.*

We note that the function $\ell_{\log}$ is 1-Lipschitz in the first argument. The result will also hold for the hinge loss $\ell_{\text{hinge}}(\hat{y},y)=(1-\hat{y}_y+\max_{y'\neq y}\hat{y}_{y'})_+$ which is also 1-Lipschitz, instead of $\ell_{\log}$.

We assume that the optimal regressor $f_{\mathcal{H}_1}^*$ for one-hot encoding also does well on linear classification.

**Assumption D.1.** *The best regressor for 1-hot encodings in $\mathcal{H}_1$ does well on classification, i.e. $\ell_{clf}(\gamma f_{\mathcal{H}_1}^*)\leq\epsilon_{\text{one-hot}}$ is small for some scalar $\gamma$.*

**Remark D.1.** *Note that if $\mathcal{H}_1$ is universal, then $f_{\mathcal{H}_1}^*(\boldsymbol{x}_1)=\mathbb{E}[Y|X_1=\boldsymbol{x}_1]$ and we know that $f_{\mathcal{H}_1}^*$ is the Bayes-optimal predictor for binary classification. In general one can potentially predict the label by looking at $\arg\max_{i\in[k]}f_{\mathcal{H}_1}^*(\boldsymbol{x}_1)_i$. The scalar $\gamma$ captures the margin in the predictor $f_{\mathcal{H}_1}^*$.*

We now show that using the classifier $\hat{W}$ obtained from linear regression on one-hot encoding with learned representations $\tilde{\psi}$ will also be good on linear classification. The proof is in Section D

**Theorem D.2.** *For a fixed $\delta \in (0, 1)$, under the same setting as Theorem 4.2 and Assumption D.1, we have:*

$$\ell_{clf}\left(\gamma \hat{W}\tilde{\psi}\right) \leq \mathcal{O}\left(\gamma\sqrt{\sigma^2\frac{d_2 + \log d_2/\delta}{n_2} + \frac{\epsilon^2}{\beta^2} + \frac{\epsilon_{pre}^2}{\beta^2}}\right) + \epsilon_{one\text{-}hot},$$

*with probability $1 - \delta$.*

**Proof of Theorem D.2.** We simply follow the following sequence of steps

$$\begin{aligned}
\ell_{\text{clf}}\left(\gamma\hat{W}\tilde{\psi}\right) &= \mathbb{E}[\ell_{\log}\left(\gamma\hat{W}\tilde{\psi}(X_1), Y\right)] \\
&\leq^{(a)} \mathbb{E}\left[\ell_{\log}\left(\gamma f_{\mathcal{H}_1}^*(X_1), Y\right) + \gamma\|\hat{W}\tilde{\psi}(X_1) - f_{\mathcal{H}_1}^*(X_1)\|\right] \\
&\leq^{(b)} \epsilon_{\text{one-hot}} + \gamma\sqrt{\mathbb{E}\left[\|\hat{W}\tilde{\psi}(X_1) - f_{\mathcal{H}_1}^*(X_1)\|^2\right]} \\
&= \epsilon_{\text{one-hot}} + \gamma\sqrt{\text{ER}_{\tilde{\psi}}[\hat{W}]}
\end{aligned}$$

where $(a)$ follows because $\ell_{\log}$ is 1-Lipschitz and $(b)$ follows from Assumption D.1 and Jensen's inequality. Plugging in Theorem 4.2 completes the proof. $\qquad\square$

# E   FOUR DIFFERENT WAYS TO USE CI

In this section we propose four different ways to use conditional independence to prove zero approximation error, i.e.,

**Claim E.1** (informal). *When conditional independence is satisfied: $X_1 \perp X_2 | Y$, and some non-degeneracy is satisfied, there exists some matrix $W$ such that $\mathbb{E}[Y|X_1] = W \, \mathbb{E}[X_2|X_1]$.*

We note that for simplicity, most of the results are presented for the jointly Gaussian case, where everything could be captured by linear conditional expectation $\mathbb{E}^L[Y|X_1]$ or the covariance matrices. When generalizing the results for other random variables, we note just replace $X_1, X_2, Y$ by $\phi_1(X_1), \phi_2(X_2), \phi_y(Y)$ will suffice the same arguments.

## E.1   INVERSE COVARIANCE MATRIX

Write $\Sigma$ as the covariance matrix for the joint distribution $P_{X_1 X_2 Y}$.

$$\Sigma = \begin{bmatrix} \Sigma_{XX} & \Sigma_{XY} \\ \Sigma_{YY}^\top & \Sigma_{YY} \end{bmatrix}, \quad \Sigma^{-1} = \begin{bmatrix} A & \rho \\ \rho^\top & B \end{bmatrix}$$

where $A \in \mathbb{R}^{(d_1+d_2)\times(d_1+d_2)}, \rho \in \mathbb{R}^{(d_1+d_2)\times k}, B \in \mathbb{R}^{k\times k}$. Furthermore

$$\rho = \begin{bmatrix} \rho_1 \\ \rho_2 \end{bmatrix}; \quad A = \begin{bmatrix} A_{11} & A_{12} \\ A_{21} & A_{22} \end{bmatrix}$$

for $\rho_i \in \mathbb{R}^{d_i \times k}, i = 1, 2$ and $A_{ij} \in \mathbb{R}^{d_i \times d_j}$ for $i, j \in \{1, 2\}$.

**Claim E.2.** *When conditional independence is satisfied, $A$ is block diagonal matrix, i.e., $A_{12}$ and $A_{21}$ are zero matrices.*

**Lemma E.3.** *We have the following*

$$\mathbb{E}[X_1|X_2] = (A_{11} - \bar{\rho}_1\bar{\rho}_1^\top)^{-1}(\bar{\rho}_1\bar{\rho}_2^\top - A_{12})X_2 \tag{17}$$

$$\mathbb{E}[X_2|X_1] = (A_{22} - \bar{\rho}_2\bar{\rho}_2^\top)^{-1}(\bar{\rho}_2\bar{\rho}_1^\top - A_{21})X_1 \tag{18}$$

$$\mathbb{E}[Y|X] = -B^{-\frac{1}{2}}(\bar{\rho}_1^\top X_1 + \bar{\rho}_2^\top X_2) \tag{19}$$

*where $\bar{\rho}_i = \rho_i \boldsymbol{B}^{-\frac{1}{2}}$ for $i \in \{1, 2\}$. Also,*

$$(\boldsymbol{A}_{11} - \bar{\rho}_1 \bar{\rho}_1^\top)^{-1} \bar{\rho}_1 \bar{\rho}_2^\top = \frac{1}{1 - \bar{\rho}_1^\top \boldsymbol{A}_{11}^{-1} \bar{\rho}_1} \boldsymbol{A}_{11}^{-1} \bar{\rho}_1 \bar{\rho}_2^\top$$

$$(\boldsymbol{A}_{22} - \bar{\rho}_2 \bar{\rho}_2^\top)^{-1} \bar{\rho}_2 \bar{\rho}_1^\top = \frac{1}{1 - \bar{\rho}_2^\top \boldsymbol{A}_{22}^{-1} \bar{\rho}_2} \boldsymbol{A}_{22}^{-1} \bar{\rho}_2 \bar{\rho}_1^\top$$

*Proof.* We know that $\mathbb{E}[X_1 | X_2] = \boldsymbol{\Sigma}_{12} \boldsymbol{\Sigma}_{22}^{-1} X_2$ and $\mathbb{E}[X_2 | X_1] = \boldsymbol{\Sigma}_{21} \boldsymbol{\Sigma}_{11}^{-1} x_1$, where

$$\boldsymbol{\Sigma}_{XX} = \begin{bmatrix} \boldsymbol{\Sigma}_{11} & \boldsymbol{\Sigma}_{12} \\ \boldsymbol{\Sigma}_{21} & \boldsymbol{\Sigma}_{22} \end{bmatrix}$$

First using $\boldsymbol{\Sigma}\boldsymbol{\Sigma}^{-1} = I$, we get the following identities

$$\boldsymbol{\Sigma}_{XX} \boldsymbol{A} + \boldsymbol{\Sigma}_{XY} \rho^\top = \boldsymbol{I} \tag{20}$$

$$\boldsymbol{\Sigma}_{XY}^\top \boldsymbol{A} + \boldsymbol{\Sigma}_{YY} \rho^\top = 0 \tag{21}$$

$$\boldsymbol{\Sigma}_{XX} \rho + \boldsymbol{\Sigma}_{XY} \boldsymbol{B} = 0 \tag{22}$$

$$\boldsymbol{\Sigma}_{XY}^\top \rho + \boldsymbol{\Sigma}_{YY} \boldsymbol{B} = \boldsymbol{I} \tag{23}$$

From Equation (22) we get that $\boldsymbol{\Sigma}_{XY} = -\boldsymbol{\Sigma}_{XX} \rho \boldsymbol{B}^{-1}$ and plugging this into Equation (20) we get

$$\boldsymbol{\Sigma}_{XX} \boldsymbol{A} - \boldsymbol{\Sigma}_{XX} \rho \boldsymbol{B}^{-1} \rho^\top = \boldsymbol{I}$$

$$\implies \boldsymbol{\Sigma}_{XX} = (\boldsymbol{A} - \rho \boldsymbol{B}^{-1} \rho^\top)^{-1} = (\boldsymbol{A} - \bar{\rho}\bar{\rho}^\top)^{-1}$$

$$\implies \begin{bmatrix} \boldsymbol{\Sigma}_{11} & \boldsymbol{\Sigma}_{12} \\ \boldsymbol{\Sigma}_{21} & \boldsymbol{\Sigma}_{22} \end{bmatrix} = \left( \begin{bmatrix} \boldsymbol{A}_{11} - \bar{\rho}_1 \bar{\rho}_1^\top & \boldsymbol{A}_{12} - \bar{\rho}_1 \bar{\rho}_2^\top \\ \boldsymbol{A}_{21} - \bar{\rho}_2 \bar{\rho}_1^\top & \boldsymbol{A}_{22} - \bar{\rho}_2 \bar{\rho}_2^\top \end{bmatrix} \right)^{-1}$$

We now make use of the following expression for inverse of a matrix that uses Schur complement: $\boldsymbol{M}/\alpha = \delta - \gamma \alpha^{-1} \beta$ is the Schur complement of $\alpha$ for $\boldsymbol{M}$ defined below

If $\boldsymbol{M} = \begin{bmatrix} \alpha & \beta \\ \gamma & \delta \end{bmatrix}$, then, $\boldsymbol{M}^{-1} = \begin{bmatrix} \alpha^{-1} + \alpha^{-1}\beta(\boldsymbol{M}/\alpha)^{-1}\gamma\alpha^{-1} & -\alpha^{-1}\beta(\boldsymbol{M}/\alpha)^{-1} \\ -(\boldsymbol{M}/\alpha)^{-1}\gamma\alpha^{-1} & (\boldsymbol{M}/\alpha)^{-1} \end{bmatrix}$

For $\boldsymbol{M} = (\boldsymbol{A} - \bar{\rho}\bar{\rho}^\top)$, we have that $\boldsymbol{\Sigma}_{XX} = \boldsymbol{M}^{-1}$ and thus

$$\boldsymbol{\Sigma}_{12} \boldsymbol{\Sigma}_{22}^{-1} = -\alpha^{-1}\beta(\boldsymbol{M}/\alpha)^{-1}((\boldsymbol{M}/\alpha)^{-1})^{-1}$$

$$= -\alpha^{-1}\beta$$

$$= (\boldsymbol{A}_{11} - \bar{\rho}_1 \bar{\rho}_1^\top)^{-1}(\bar{\rho}_1 \bar{\rho}_2^\top - \boldsymbol{A}_{12})$$

This proves Equation (17) and similarly Equation (18) can be proved.

For Equation (19), we know that $\mathbb{E}[Y | X = (X_1, X_2)] = \boldsymbol{\Sigma}_{YX} \boldsymbol{\Sigma}_{XX}^{-1} X = \boldsymbol{\Sigma}_{XY}^\top \boldsymbol{\Sigma}_{XX}^{-1} X$. By using Equation (22) we get $\boldsymbol{\Sigma}_{XY} = -\boldsymbol{\Sigma}_{XX} \rho \boldsymbol{B}^{-1}$ and thus

$$\mathbb{E}[Y | X = (X_1, X_2)] = -\boldsymbol{B}^{-1} \rho^\top \boldsymbol{\Sigma}_{XX} \boldsymbol{\Sigma}_{XX}^{-1} X$$

$$= -\boldsymbol{B}^{-1} \rho^\top X = \boldsymbol{B}^{-1}(\rho_1^\top X_1 + \rho_2^\top X_2)$$

$$= -\boldsymbol{B}^{-\frac{1}{2}}(\bar{\rho}_1^\top X_1 + \bar{\rho}_2^\top X_2)$$

For the second part, we will use the fact that $(\boldsymbol{I} - \boldsymbol{a}\boldsymbol{b}^\top)^{-1} = \boldsymbol{I} + \frac{1}{1 - \boldsymbol{a}^\top \boldsymbol{b}} \boldsymbol{a}\boldsymbol{b}^\top$. Thus

$$(\boldsymbol{A}_{11} - \bar{\rho}_1 \bar{\rho}_1^\top)^{-1} \bar{\rho}_1 \bar{\rho}_2 = (\boldsymbol{I} - \boldsymbol{A}_{11}^{-1} \bar{\rho}_1 \bar{\rho}_1^\top) \boldsymbol{A}_{11}^{-1} \bar{\rho}_1 \bar{\rho}_2^\top$$

$$= (\boldsymbol{I} + \frac{1}{1 - \bar{\rho}_1^\top \boldsymbol{A}_{11}^{-1} \bar{\rho}_1} \boldsymbol{A}_{11}^{-1} \bar{\rho}_1 \bar{\rho}_1) \boldsymbol{A}_{11}^{-1} \bar{\rho}_1 \bar{\rho}_2^\top$$

$$= \boldsymbol{A}_{11}^{-1}(\boldsymbol{I} + \frac{1}{1 - \bar{\rho}_1^\top \boldsymbol{A}_{11}^{-1} \bar{\rho}_1} \bar{\rho}_1 \bar{\rho}_1 \boldsymbol{A}_{11}^{-1}) \bar{\rho}_1 \bar{\rho}_2^\top$$

$$= \boldsymbol{A}_{11}^{-1}(\bar{\rho}_1 \bar{\rho}_2^\top + \frac{\bar{\rho}_1 \boldsymbol{A}_{11}^{-1} \bar{\rho}_1}{1 - \bar{\rho}_1^\top \boldsymbol{A}_{11}^{-1} \bar{\rho}_1} \bar{\rho}_1 \bar{\rho}_2^\top)$$

$$= \boldsymbol{A}_{11}^{-1} \bar{\rho}_1 \bar{\rho}_2^\top (1 + \frac{\bar{\rho}_1 \boldsymbol{A}_{11}^{-1} \bar{\rho}_1}{1 - \bar{\rho}_1^\top \boldsymbol{A}_{11}^{-1} \bar{\rho}_1})$$

$$= \frac{1}{1 - \bar{\rho}_1^\top \boldsymbol{A}_{11}^{-1} \bar{\rho}_1} \boldsymbol{A}_{11}^{-1} \bar{\rho}_1 \bar{\rho}_2^\top$$

The other statement can be proved similarly. □

**Claim E.4.**

$$\mathbb{E}[X_2|X_1] = (\boldsymbol{A}_{22} - \bar{\rho}_2 \bar{\rho}_2^\top)^{-1} \bar{\rho}_2 \bar{\rho}_1^\top X_1. \mathbb{E}[Y|X_1] = -\boldsymbol{B}^{-1/2} \bar{\rho}_1^\top X_1 - \boldsymbol{B}^{-1/2} \bar{\rho}_2^\top \mathbb{E}[X_2|X_1]$$

*Therefore* $\mathbb{E}[Y|X_1]$ *is in the same direction as* $\mathbb{E}[X_2|X_1]$.

### E.2 Closed form of Linear Conditional Expectation

Refer to Claim 3.1 and proof of Lemma 3.2. As this is the simplest proof we used in our paper.

### E.3 From Law of Iterated Expectation

$$\mathbb{E}^L[X_2|X_1] = \mathbb{E}^L[\mathbb{E}^L[X_2|X_1, Y]|X_1]$$
$$= \mathbb{E}\left[ [\boldsymbol{\Sigma}_{X_2 X_1}, \boldsymbol{\Sigma}_{X_2 Y}] \begin{bmatrix} \boldsymbol{\Sigma}_{X_1 X_1} & \boldsymbol{\Sigma}_{X_1 Y} \\ \boldsymbol{\Sigma}_{Y X_1} & \boldsymbol{\Sigma}_{YY} \end{bmatrix}^{-1} \begin{bmatrix} X_1 \\ Y \end{bmatrix} \mid X_1 \right]$$
$$= \boldsymbol{A} X_1 + \boldsymbol{B} \, \mathbb{E}^L[Y|X_1].$$

Using block matrix inverse,

$$\boldsymbol{A} = (\boldsymbol{\Sigma}_{X_2 X_1} - \boldsymbol{\Sigma}_{X_2 Y} \boldsymbol{\Sigma}_{YY}^{-1} \boldsymbol{\Sigma}_{Y X_1})(\boldsymbol{\Sigma}_{X_1 X_1} - \boldsymbol{\Sigma}_{X_1 Y} \boldsymbol{\Sigma}_{YY}^{-1} \boldsymbol{\Sigma}_{Y X_1})^{-1} \in \mathbb{R}^{d_2 \times d_1}$$
$$= \boldsymbol{\Sigma}_{X_1 X_2|Y} (\boldsymbol{\Sigma}_{X_1 X_1|Y})^{-1}$$
$$\boldsymbol{B} = \boldsymbol{\Sigma}_{X_2 Y|X_1} (\boldsymbol{\Sigma}_{YY|X_1})^{-1} \in \mathbb{R}^{d_2 \times \mathcal{Y}}.$$

Therefore in general (without conditional independence assumption) our learned representation will be $\psi(x_1) = \boldsymbol{A} x_1 + \boldsymbol{B} f^*(x_1)$, where $f^*(\cdot) := \mathbb{E}^L[Y|X_1]$.

It's easy to see that to learn $f^*$ from representation $\psi$, we need $A$ to have some good property, such as light tail in eigenspace, and $B$ needs to be full rank in its column space.

Notice in the case of conditional independence, $\boldsymbol{\Sigma}_{X_1 X_2|Y} = 0$, and $A = 0$. Therefore we could easily learn $f^*$ from $\psi$ if $X_2$ has enough information of $Y$ such that $\boldsymbol{\Sigma}_{X_2 Y|X_1}$ is of the same rank as dimension of $Y$.

### E.4 From $\mathbb{E}[X_2|X_1, Y] = \mathbb{E}[X_2|Y]$

*Proof.* Let the representation function $\psi$ be defined as follows, and let we use law of iterated expectation:

$$\psi(\cdot) := \mathbb{E}[X_2|X_1] = \mathbb{E}[\mathbb{E}[X_2|X_1, Y]|X_1]$$
$$= \mathbb{E}[\mathbb{E}[X_2|Y]|X_1] \qquad \text{(uses CI)}$$
$$= \sum_y P(Y = y|X_1) \, \mathbb{E}[X_2|Y = y]$$
$$=: f(X_1)^\top A,$$

where $f : \mathbb{R}^{d_1} \to \Delta_{\mathcal{Y}}$ satisfies $f(x_1)_y = P(Y = y|X_1 = x_1)$, and $\boldsymbol{A} \in \mathbb{R}^{\mathcal{Y} \times d_2}$ satisfies $\boldsymbol{A}_{y,:} = \mathbb{E}[X_2|Y = y]$. Here $\Delta_d$ denotes simplex of dimension $d$, which represents the discrete probability density over support of size $d$.

Let $\boldsymbol{B} = \boldsymbol{A}^\dagger \in \mathbb{R}^{\mathcal{Y} \times d_2}$ be the pseudoinverse of matrix $\boldsymbol{A}$, and we get $\boldsymbol{B} \boldsymbol{A} = \boldsymbol{I}$ from our assumption that $A$ is of rank $|\mathcal{Y}|$. Therefore $f(\boldsymbol{x}_1) = \boldsymbol{B} \psi(\boldsymbol{x}_1), \forall x_1$. Next we have:

$$\mathbb{E}[Y|X_1 = \boldsymbol{x}_1] = \sum_y P(Y = y|X_1 = \boldsymbol{x}_1) \times y$$
$$= \hat{\boldsymbol{Y}} f(\boldsymbol{x}_1)$$
$$= (\hat{\boldsymbol{Y}} \boldsymbol{B}) \cdot \psi(X_1).$$

Here we denote by $\hat{\boldsymbol{Y}} \in \mathbb{R}^{k \times \mathcal{Y}}, \hat{\boldsymbol{Y}}_{:,y} = y$ that spans the whole support $\mathcal{Y}$. Therefore let $\boldsymbol{W}^* = \hat{\boldsymbol{Y}} \boldsymbol{B}$ will finish the proof.

$\square$

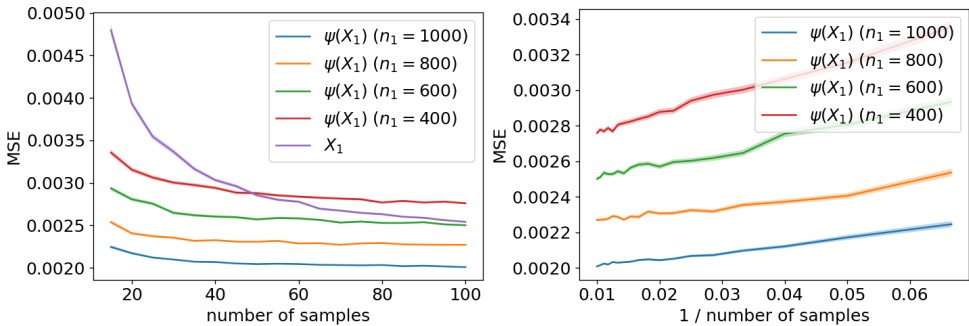

Figure 3: **Left**: MSE of using $\psi$ to predict $Y$ versus using $X_1$ directly to predict $Y$. Using $\psi$ consistently outperforms using $X_1$. **Right**: MSE of $\psi$ learned with different $n_1$. The MSE scale with $1/n_2$ as indicated by our analysis. Simulations are repeated 100 times, with the mean shown in solid line and one standard error shown in shadow.

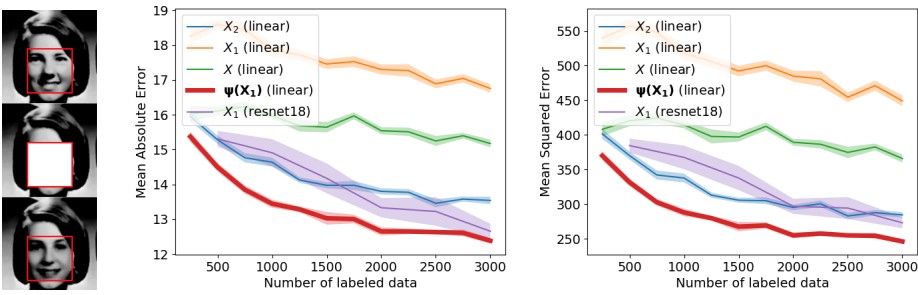

Figure 4: **Left**: Example of the $X_2$ (in the red box of the 1st row), the $X_1$ (out of the red box of the 1st row), the input to the inpainting task (the second row), $\psi(X_1)$ (the 3 row in the red box), and in this example $Y = 1967$. **Middle**: Mean Squared Error comparison of yearbook regression predicting dates. **Right**: Mean Absolute Error comparison of yearbook regression predicting dates. Experiments are repeated 10 times, with the mean shown in solid line and one standard error shown in shadow.

## F    MORE ON THE EXPERIMENTS

In this section, we describe more experiment results.

**Simulations.**    Following Theorem 4.2, we know that the Excessive Risk (ER) is also controlled by (1) the number of samples for the pretext task ($n_1$), and (2) the number of samples for the downstream task ($n_2$), besides $k$ and $\epsilon_{CI}$ as discussed in the main text. In this simulation, we enforce strict conditional independence, and explore how ER varies with $n_1$ and $n_2$. We generate the data the same way as in the main text, and keep $\alpha = 0, k = 2, d_1 = 50$ and $d_2 = 40$ We restrict the function class to linear model. Hence $\psi$ is the linear model to predict $X_2$ from $X_1$ given the pretext dataset. We use Mean Squared Error (MSE) as the metric, since it is the empirical version of the ER. As shown in Figure 3, $\psi$ consistently outperforms $X_1$ in predicting $Y$ using a linear model learnt from the given downstream dataset, and ER does scale linearly with $1/n_2$, as indicated by our analysis.

**Computer Vision Task.**    We testify if learning from $\psi$ is more effective than learning directly from $X_1$, in a realistic setting (without enforcing conditional independence). Specifically, we test on the Yearbook dataset (Ginosar et al., 2015), and try to predict the date when the portraits are taken (denoted as $Y_D$), which ranges from 1905 to 2013. We resize all the portraits to be 128 by 128. We crop out the center 64 by 64 pixels (the face), and treat it as $X_2$, and treat the outer rim as $X_1$ as shown in Figure 4. Our task is to predict $Y_D$, which is the year when the portraits are taken, and the year ranges from 1905 to 2013. For $\psi$, we learn $X_2$ from $X_1$ with standard image inpainting

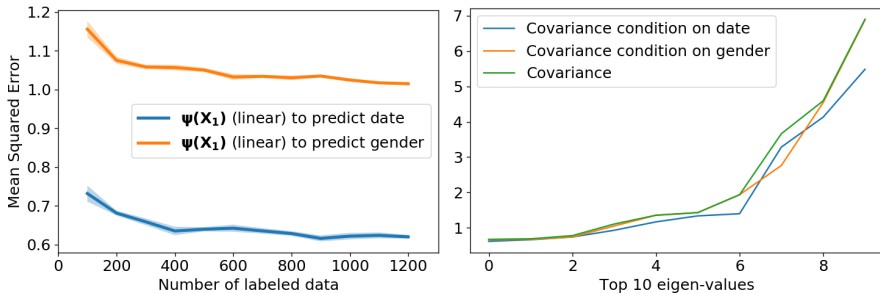

Figure 5: **Left**: Mean Squared Error comparison of predicting gender and predicting date. **Right**: the spectrum comparison of covariance condition on gender and condition on date.

techniques (Pathak et al., 2016), and full set of training data (without labels). After that we fix the learned $\psi$ and learn a linear model to predict $Y_D$ from $\psi$ using a smaller set of data (with labels). Besides linear model on $X_1$, another strong baseline that we compare with is using ResNet18 (He et al., 2016) to predict $Y_D$ from $X_1$. With the full set of training data, this model is able to achieve a Mean Absolute Difference of $6.89$, close to what state-of-the-art can achieve (Ginosar et al., 2015). ResNet18 has similar amount of parameters as our generator, and hence roughly in the same function class. We show the MSE result as in Figure 4. Learning from $\psi$ is more effective than learning from $X_1$ or $X_2$ directly, with linear model as well as with ResNet18. Practitioner usually fine-tune $\psi$ with the downstream task, which usually leads to more competitive performance (Pathak et al., 2016).

Following the same procedure, we try to predict the gender $Y_G$. We normalize the label $(Y_G, Y_D)$ to unit variance, and confine ourself to linear function class. That is, instead of using a context encoder to impaint $X_2$ from $X_1$, we confine $\psi$ to be a linear function. As shown on the left of Figure 5, the MSE of predicting gender is higher than predicting dates. We find that $\|\Sigma_{\boldsymbol{X}_1\boldsymbol{X}_1}^{-1/2}\Sigma_{\boldsymbol{X}_1 X_2|Y_G}\|_F = 9.32$, while $\|\Sigma_{\boldsymbol{X}_1\boldsymbol{X}_1}^{-1/2}\Sigma_{\boldsymbol{X}_1 X_2|Y_D}\|_F = 8.15$. Moreover, as shown on the right of Figure 5, conditioning on $Y_D$ cancels out more spectrum than conditioning on $Y_G$. In this case, we conjecture that, unlike $Y_D$, $Y_G$ does not capture much dependence between $X_1$ and $X_2$. And as a result, $\epsilon_{CI}$ is larger, and the downstream performance is worse, as we expected.

