# OpenReview forum: "Predicting What You Already Know Helps: Provable Self-Supervised Learning"
_ICLR.cc/2021/Conference — Reject_

### Official Review · AnonReviewer1 · 2020-10-27
**An interesting paper with some issues on clarify and generality**

**Rating:** 6
**Confidence:** 3

**Review:**

This paper attempts to understand why self-supervised learning works in the following sense: will the sample complexity for a downstream task be decreased (compared to the standard supervised learning without pretraining) if it is pre-trained according to some related auxiliary task? The relation between tasks is formulated as the approximate conditional independence of the dependent variables. The main theoretical results show that the sample complexity (compared to supervised learning) can be decreased under Assumption 3.5, 4.1, and 4.2.

Clarity:

Generally, this paper is well organized. I have some comments to improve clarity.

i) I'm not quite sure about the motivation of the conditional independence assumption in the introduction. In the colorization example used in the introduction, an algorithm with a general hypothesis (not CNNs) does not necessarily know the semantics of the images (even implicitly) to predict the background-color. It can focus on the cross channel information at the image border to make predictions and ignore the image semantics. Such a self-supervised learning method works (i.e., extracts semantic features) may simply because of the inductive bias introduced by the hypothesis (i.e., CNNs). Therefore, I can hardly say that the pretext tasks work because the "only" way is to implicitly infer Y first given the conditional independence assumption.

ii) It may be better to explain more about the assumptions and conclusions (lemmas and thms) considering both the theory and practice in self-supervised learning.

For instance, about theorem 4.2, the paper only mentions that the bound is independent of the complexity of the feature extractor. Why can you achieve this? I guess it is because of the two-stage training and a strong assumption on the feature extractor. Any technical contributions to achieve this should be highlighted? Do the results of traditional supervised learning rely on a similar assumption? Besides, around theorem 4.2, I didn't see a sample complexity comparison to supervised learning. Maybe it is well known, but this is the main claim of the paper.

Further, I wonder how good the sample complexity is, compared to the results in some related areas like transfer learning and semi-supervised learning (not just supervised learning). Self-supervised learning has outperformed other methods in transfer learning and semi-supervised learning and it would be better to at least discuss more.

Correctness:

I'm not an expert in machine learning theory. I read the main text carefully and I understood the setups, assumptions, and main results. I quickly checked the proof in the appendix and didn't find flaws. I'm aware that I may underestimate the theoretical contribution of the paper since I'm not familiar with the related work. Welcome any feedback on this during rebuttal and discussion.

As for empirical validation, considering that the main result is Theorem 4.2 and the main "baseline" is supervised learning, it would be better to directly compare the sample complexities of supervised learning and self-supervised learning in experiments.

Generality:

One major concern is that the conditional independence assumption (even the approximate one) can be violated in practice. For instance, recently, the instance discrimination pretext has been extensively studied in self-supervised learning and achieved SOTA results in many benchmarks  (e.g., in MoCo and many recent papers). In that case, the random variable X2 is the index of the image (or the input generally). Since we can shuffle the index, it seems that we cannot say X2 (index) and X1 (image) are (approximately) conditionally independent conditioned on Y(label). Also, the paper mentions that some concurrent work does not apply to recontrsuction SSL in the related work. I think, in this case,  X1 = X2, and the conditional independence assumption does not hold as well. In such cases, does the analysis in this paper still apply? The generality of the analysis should be discussed somewhere in the paper because i) the title seems to cover all self-supervised learning methods and ii) the main difference of the paper from several recent papers is the generality as claimed in the related work.

Another issue is that the paper discusses two hypotheses H1 and H_u and also mentions that H_u includes deep neural networks. However, as far as I know, only infinitely wide or deep neural networks are in H_u.  In practice, self-supervised learning often uses networks with finite depth and width. Does the analysis in this paper apply in this case? If so, please also add detailed discussion and some validation experiments in such models. Otherwise, please clarify this since it can be misleading.

Related work:

Compared to the existing work, the main contribution of the paper is to generalize the conditional independence based analysis via weakening the assumption. We may need to pay attention to that some concurrent work mentioned in this paper obtain similar results (with less generality as claimed in the related work but also see my comments in Generality).

Overall, I think the theory of self-supervised learning is very important and I should not blame the authors too much since it is an early work on this topic. However, I hope the authors to address the above issues to improve paper quality. I give a rating of 6 currently but it is very borderline in my opinion.

---

> ### Author Response · Authors · 2020-11-20
> **Response to Reviewer 1 (1/2)**
>
> We thank the reviewer for the insightful suggestions. We will try to add more clarifications to our paper.
>
> **[I can hardly say that the pretext tasks work because the "only" way is to implicitly infer Y first]**
> A correct rephrasing would be "Under CI assumption, predicting $X_2$ from $X_1$ will always implicitly encode and learn to predict $Y$ from $X_1$ as an intermediate step" and we will fix this.
> We do not actually need the know the label $Y$ during the pre-training phase, but if ACI holds, the predicted value of $X_2$ will automatically encode $Y$ linearly (which we have proved).
> Note that the additional latent variables in the definition of ACI can capture some small dependencies of $X_2$ on $X_1$ that are not related to $Y$, thus allowing for usage of "cross channel information".
> Furthermore, there might be also be some implicit dependencies of the cross channel information on the label itself, which will further reduce the ACI parameter $\epsilon_{CI}$.
>
> **[Clarity on Theorem 4.2, sample complexity of supervised learning]**
> Our target is to learn a predictor function of the form $f(x)=A\psi(x)$, i.e, a feature extractor $\psi\in\mathcal{H}$ (which could be viewed as a neural network without the last layer) composed with a linear mapping. In supervised learning, $f$ is entirely learned using labeled data and the sample complexity is known to be upper bounded by some complexity measures of the function class of $f$, (usually measured by Gaussian width or Rademacher complexity) which is larger than the complexity of $\mathcal{H}$. We discuss this below Theorem 3.9. However we do not report the exact bound because it depends on the specific choice of $\mathcal{H}$. However for most $\mathcal{H}$ of interest, the sample complexity of learning it is much higher than learning only a linear mapping as in our setup.
>
> The independence on complexity of feature extractor is precisely because of the two-stage process, where the first self-supervised stage learn a complicated representation function ($\psi:X_1\rightarrow \mathbb{R}^{d_2}$) by using an abundance of *unlabeled data* $X_1,X_2$, while the second stage just needs to learn a simple linear classifier using few labeled data. Our paper proves that with only unlabeled data, the first stage is able to find a good representation $\psi$ such that the best prediction function for $Y$ is linear in $\psi(X_1)$. Therefore in order to find the best predictor $f$, we only need at most $O(d_2)$ (or $O(k)$ with exact CI) labeled data to learn the last linear layer. We will make this point clearer in the revision.
>
> **[wonder how good the sample complexity is compare to transfer learning and semi-supervised learning]**
> It is known that in order to learn a linear mapping, the sample complexity is lower bounded by the data dimension $d_2$. For instance, the minimax lower bound for $d-$dimensional linear regression is $O(d/n)$ in (7.4.13) in book [1]. Therefore our sample complexity is tight up to a constant factor.
>
> With exact CI we are able to learn $Y$ with $O(k)$ samples, which is information theoretically optimal considering $Y$ has dimension $k$.
>
> Our empirical findings on the risk being linear in $\epsilon_{CI}$ is also a side proof that the corresponding term in our upper bound is not reducible.
>
> Precise mathematical comparisons should be conducted under similar settings. Prior work on semi-supervised or transfer learning is conducted in very different settings and might not be comparable to this paper.
>
> **[Directly compare the sample complexities of supervised learning and self-supervised learning in experiments]**
> Please refer to Figure 3 and Figure 4 in the Appendix for direct comparisons of sample complexities of SSL and supervised learning. In both the simulations and CV tasks, we use exactly the same setting described in our theory as the SSL algorithm and observe that supervised learning requires much higher samples than ours to achieve a similar performance.

---

> > ### Author Response · Authors · 2020-11-20
> > **Response to Reviewer 1 (2/2)**
> >
> > **[Generality of the analysis should be discussed. Title seems to cover all self-supervised learning methods]**
> > Firstly we would like to note that our title does not imply that we cover all SSL methods. The first part "Predicting What You Already Know Helps" already places emphasis on reconstruction-based methods (predicting part of an input from the rest) as opposed to methods like contrastive learning that aim for representation alignment. The second part "Provable Self-Supervised Learning" implies provable guarantees for *one* SSL method that is motivated by reconstruction-based SSL. Indeed it would be misleading (and ever over-ambitious) to claim provable guarantees for all SSL methods and we do not intend to do so.
> > As you point out, ACI does not hold for the instance discrimination pretext task for the obvious reason and thus our results won't give a meaningful upper bound. The main difference from prior work is our generality in the assumption (ACI with latent variables as opposed to exact CI with labels), but not generality in the SSL method considered.
> >
> > **[Does the analysis apply to non-universal neural networks. Provide experimental validation for such models]**
> > Our most general result from Theorem 4.2 holds for any representation function $\tilde{\psi}$ that is $\epsilon^2_{pre}$-optimal in the pretext task, regardless of the function class that is used.
> > Using a non-universal function class does not really matter, as long as it is able to make $\epsilon^2_{pre}$ small. The theory does not need infinite-width networks. Our experiments for the computer vision task in Section F use finite-width neural networks for the SSL phase and do well on the downstream task.

---

### Official Review · AnonReviewer3 · 2020-10-27
**Review of "Predicting What You Already Know Helps: Provable Self-Supervised Learning".**

**Rating:** 6
**Confidence:** 3

**Review:**



  *  Summary of the paper.
  The paper shows theoretical results to support the claim that (approximate) conditional independence is a good way to quantify the link between the downstream task and the pretext task in self-supervised learning. Doing so, the authors prove theoretical results showing that indeed self-supervised learning decrease the estimation error in a classification task when the downstream task and the pretext task are linked through conditional independence or the weaker approximate conditional independence showing that approximate conditional independence is a good quantification of the link that the downstream task and the pretext task must have in order for the self-supervised learning to be efficient. The authors also provide numerical illustrations.

  * Strong points: theoretical guarantees in SSL (there are not a lot of those) while using approximate conditional independence which is more realistic than conditional independence.  The paper has some experiments that support its claim. The paper takes a gradual approach, first with jointly Gaussian variables and then with general (sub-Gaussian) random variables which helps the comprehension because at least in the Gaussian case, things are not so hard.

  * Weak points: Use of mean squared error for a classification task which negates most of the practical aspect of this paper. Not enough details in the proofs. The bounds are not optimal. The experiments are not reproducible and the presentation of those experiments is somewhat lacking. Overall problems with experiments.

  * Recommendation.
    I vote to accept because this is one of the rare theoretical guarantees for SSL, the approximate conditional independence is very interesting as a way to quantify the link between downstream and pretext task and this gives more understanding on how to choose a pretext task in practice when doing SSL.


  *  Questions:
    * Why did you use MSE ? Why not Cross entropy for instance ?
    * $Tr(\Sigma_{YY})$ in Theorem 3.3 is of order k for instance in the case where $\Sigma_{YY}$ is the identity matrix, the bound is $O(k^2)$ not $O(k)$, is it not? (contradiction with what you say in the text).


  * Additional feedback.
    * Please provide the minimax bound (i.e. optimal bound) in the "not self-supervised" context for us to compare with your bounds for self-supervised, below Theorem 3.3 you say that we gain from $O(d1)$ to $O(k)$ but you don't provide the bound that explain this $O(d1)$.
    * Please include the results of the simulation study in the main text. In my opinion, a simulation study must firstly show that your method works as intended, a sanity check of sorts. You simulation study does not do this because we don't see the results of this study right away, it is hidden at the end of the appendix.
    * Be careful, Theorem A.6 is only true for sub-Gaussian, please include this in the theorem, this is important in my opinion.
    * Theorem A.6 can be improved, for now it is of order $\sqrt{\frac{Tr(\Sigma)}{n}}(1+t)$ but it can be made of order
$\sqrt{\frac{Tr(\Sigma)}{n}}+\sqrt{\frac{||\Sigma||_{op}t}{n}}.$

        The difference can be important when the dimension is large (for instance if Sigma is the identity matrix in $R^d$, $Tr(\Sigma)= d$, $||\Sigma||_{op}=1$). To obtain this for sub-Gaussian vectors, one can for instance use the article "A tail inequality for suprema of unbounded empirical processes with applications to Markov chains" by R. Adamczak. This is important because the dimension is important for your conclusion.
    * Lemma A.7 is not true for any delta, please state in the lemma for which delta it is true (delta is at most $ke^{-n}$ if I am not wrong)
    * Generally, when reading the proofs I would have appreciated if you included more details. The "therefore" in the proof of theorem 3.3 was not clear right away for me, there are numerous lines where you do a lot of manipulations that the reader has to guess to obtain the good results and this makes it very hard for us to check the results you claim.
    * In the proof of Lemma C.2, please provide a reference for "Davis kahan", for those that are not familiar with this result.
    * Typo: beginning of the first sentence of A1.

---

> ### Author Response · Authors · 2020-11-20
> **Response to Reviewer 3**
>
> Thank you for the detailed feedback and useful suggestions.
>
> **[Why use MSE for downstream tasks?]**
> In the main text, our analysis covers ordinal regression and therefore we use square loss. For classification tasks, there is strong empirical evidence [1] that using square loss performs comparably to cross-entropy on many classification tasks, thus justifying the use of the square error in our theory.
> Furthermore, Section D provides an analysis of the cross-entropy loss for classification tasks as well.
>
>
> **[$Tr(\Sigma_{YY})$ makes the bound $k^2$ instead of $k$]**
> Our main result is when $Y$ is categorical (where Tr$(\Sigma_{YY})$ is $O(1)$) and therefore our bound is indeed $\tilde{O}(k)$. Regardless, the improvement can always be viewed as a multiplicative factor of $\frac{d}{k}$ in this section.
>
> **[Do not provide a lower bound of $\mathcal{O}(d_1)$]**
> For instance, the minimax lower bound for $d-$dimensional linear regression is $O(d/n)$ in (7.4.13) in book [2].
>
> **[Please include the results of the simulation study in the main text]**
> We have put most implementation details in the appendix. We will make the main text clearer in the revised version.
>
> **[Theorem A.6 and Lemma A.7 can be improved]**
> Thank you for your suggestions. However, in our main results, we are considering the case where each label is categorical and therefore the covariance of $Y$ has a trace of $O(1)$ and not dependent on the dimension $k$. We don't see much room for improvement except for some log factors. We will add the range for $\delta$ in Lemma A.7; thank you for pointing out.
>
> We will fix citation, typos, and add more details in the proofs.
>
>
> [1] Like Hui, Mikhail Belkin; Evaluation of Neural Architectures Trained with Square Loss vs Cross-Entropy in Classification Tasks
>
> [2] Duchi, John. "Lecture notes for statistics 311/electrical engineering 377.".

---

### Official Review · AnonReviewer4 · 2020-10-28
**Interesting topic and useful findings**

**Rating:** 6
**Confidence:** 5

**Review:**

This paper proposes a mechanism based on approximate conditional independence (ACI) to explain why solving pretext tasks created from known information can learn representations that provably reduce downstream sample complexity, as a sufficient condition. In specific, they measure the downstream performance using the approximation error and estimation error, and establish their initial results under a strict condition -- conditional independence (CI) and linear function space in Section 3, then extend it to ACI and arbitrary function space in Section 4, resulting in the main contribution Theorem 4.2 that clearly quantifies the generalization error. The theorem is also verified in Section 5 using simulations on NLP tasks.

One concern is that, in terms of the theory for self-supervised learning, I am not sure how big is the proposed theorem contributes to the community, as there are works such as Tosh et al. (2020b) in contrastive learning also provide theoretical guarantees using assumptions similar to CI. So I would rather hear more comments from other reviewers.

Another point is that the authors spent around four pages to illustrate their results with lemmas and theorems, which is good. But for the benefit of a broader audience, I would suggest to include more intuitions and discussions at the beginning, include some proof steps and ideas only with limited but key lemmas and theorems, and move the rest of them into the appendix. For example, Section 3.1 is on jointly Gaussian variables and 3.2 is on general random variables, they are indeed different, but results are pretty same and kind of overlap with each other. I would suggest shrink them but add more discussions and intuitions for the audience.

Overall, from my own point of view, the paper is clearly written and well-motivated. It addresses an important question that perfectly fits into the ICLR topics: what connection between pretext and downstream tasks ensures good representations?


------
Update after rebuttal

Thanks for the author's response. Combing the author's response (though the authors didn't upload new versions to address my comments -- include more intuitions and discussions) and other reviewer's comments and discussions. I suggested this paper being marginally above the acceptance threshold.

---

> ### Author Response · Authors · 2020-11-20
> **Response to Reviewer 4**
>
> Thank you for the suggestions. We will try to add more intuitions and discussions in the main paper. We answer your main concern below.
>
> **[Contribution to theory of SSL]**
> We would like to point out that Tosh et al. (2020b) was released on arxiv on Aug 24, 2020  and anything after Aug 2, 2020 is considered contemporaneous work  based on ICLR 2021 FAQs. That being said, Tosh et al. (2020b) provides guarantees for contrastive-based SSL rather than reconstruction-based SSL under a different looking, but very related assumption.
> Thus their work is complementary to ours, and both works exploit CI-like assumptions to show the benefit of two different SSL methods.
> We believe that the presence of their work does not dilute the contributions of our paper.

---

### Official Review · AnonReviewer2 · 2020-10-29
**This is a good submission that provides some theoretical understanding of the reconstruction-based SSL.**

**Rating:** 6
**Confidence:** 3

**Review:**

The estimation error refers to the distance between the best function in some function class H, and the optimal estimator computed based on the given data. The definition at the end of page 3 may have some typos since f^* is the universal optimal predictor.
Lemma 3.5 seems trivial since \psi comes from the universal function class; it is not surprised to get the zero approximation error.
Some CI results may inspire this utilized ACI; however, the provided generalization bound seems more general with a weaker assumption.
Overall, it is a good sunmision and offers much insight for SSL from the theoretical perspective.

---

> ### Author Response · Authors · 2020-11-20
> **Response to Reviewer 2**
>
> Thank you for the suggestions and positive feedback.
>
> **[Estimation error definition]**
> We agree and will simply use excess risk instead.
>
> **[Lemma 3.5 is trivial]**
> The apx$(\psi^*)$ term refers to the approximation error on learning $Y$ with a linear function on $\psi^*(X_1)$. Even though $\psi^*$ is from a universal function class, it is not trivial that $\psi^*(X_1)$ ($=\mathbb{E}[X_2|X_1] $) can learn $f^*(X_1)=\mathbb{E}[Y|X_1]$ as a linear function. In general, this approximation error is not necessarily zero if Assumption 3.4 is not satisfied. We will make our definition of this approximation error clearer in the revised version to avoid future confusion.

---

### Official Review · AnonReviewer5 · 2020-11-04
**Is the framework really about self-supervision?**

**Rating:** 6
**Confidence:** 3

**Review:**

__post-rebuttal__
The responses have been persuasive enough. I am raising my score, with an expectation that the authors will make additional textual revisions based on their responses to make it clear in the abstract and introduction that (1) authors only consider the "reconstruction-based" SSL, instead of SSL in general, and (2) address the discrepancy between the practice and the proposed framework. (I thought asking questions could make the authors revise the manuscript, but unfortunately that did not happen.)

__summary of the paper__
The paper provides a mathematical framework to theoretically understand and quantify the benefit of self-supervision on the downstream tasks. The considered pipeline (which seems to be based on Arora et al. (2019)) is as follows. $X_1$ is the input variable, $X_2$ is the "target" random variable, and $Y$ is the label for the downstream task. Then, the learner solves two problems sequentially: (1) _pretext task:_ The learner looks for a representation $\psi: \mathcal{X}_1 \to \mathcal{X}_2$ such that $\psi(X_1) \approx X_2$. (2) _downstream task:_: The learner looks for a matrix $W$ such that $W^\top \psi(X_1) \approx Y$. Based on this framework, the paper gives performance guarantees on the downstream task under various assumptions, e.g. unbounded sample for the pretext task or Gaussianity. The key underlying mechanism is what authors call the _approximate conditional independence._

__Strengths__
(1) The theoretical analyses appearing in the paper are concrete.
(2) The manuscript overall is very easy to understand and comprehensive.

__Weakness__
The biggest concern that I have is that the provided framework does not seem to incorporate most existing self-supervised learning paradigms. Although it is somewhat explicitly stated already in the paper that the discussion is confined to the **"reconstruction-based self-supervised learning"** instead of any self-supervised learning task, I still am not really sure if the provided framework covers those pretext tasks. Actually, the paper exemplifies the task of "inpainting" (Pathak et al. 2016) to motivate the theoretical framework. Here, $X_2$ is the cropped part of an image, $X_1$ is the remaining part, and $Y$ is the corresponding label. I am happy with how the framework handles the pretext task, but not with how it handles the downstream task. According to the framework, the learner aims to predict the label from the "remainder of a cropped image," instead of the full image. While one can always imagine such an algorithm, I am not sure if this is what is typically done in self-supervision algorithms. The same may hold for image colorization tasks (which also motivates the framework). Actually, it seems to me that such discrepancy takes place because the paper did not want to step away too far from Arora et al. (2019), which is actually quite similar in spirit except that Arora et al. (2019) did not spell out "approximate conditional independence."

Also, I am not sure if the framework is extendable enough to take care of other (perhaps more popular and well-performing) categories of self-supervision, such as rotation/jigsaw-based methods. If not, I think many statements and expressions appearing in the paper should be significantly down-toned, including the title of the paper.

__nitpicks__
(1) Although not absolutely mandatory, discussing the relationship to Bansal et al. (2020) may help the future readers.
(2) In section 2, right before the paragraph **(Partial) covariance matrix,** the sentence "We also note that $\mathbb{E}[Y|X] = \min_{f}\mathbb{E}[\|Y-f(X)\|^2]$ is the best predictor of $Y$ given $X$" does not make too much sense. Perhaps the authors meant argmin?
(3) Is $X$ the random variable, or can it be a vector as well?
(4) In section 2.2., I am not sure why the authors use the Frobenius norm in the definition of  $\psi$. Shouldn't it be simply $\ell_2$ norm, as $X_2$ seems to be a vector? Or are the authors already assuming a data matrix?


[Bansal et al. 2020] For self-supervised learning, Rationality implies generalization, provably, arXiv 2020.

---

> ### Author Response · Authors · 2020-11-20
> **Response to Reviewer 5**
>
> Thank you for your insightful suggestions and for pointing out the unclear part of our contributions.
>
>
> **[Our results are based on Arora et al.]** We argue that our results are substantially different from Arora et al 2019. The algorithms, settings, assumptions, the underlying reasons for the models' success, and the proof techniques are all different. The similarities are in the standard two-stage representation learning setup and that both our methods work when CI is satisfied. The results in Arora et al., however, do not hold with approximate CI nor CI with latent variables. The basis for their argument is that contrastive learning implicitly learns a mean classifier by mapping the data samples from the same class to be close together when they are sampled from the same class. Our results focus on reconstruction-based SSL instead and suggest that when $X_1$ and $X_2$ are approximately conditional independent given finite latent variables, the prediction from $X_1$ to $X_2$ will have to encode the information of $Y$ and the latent variables, and therefore it will help with downstream tasks. We hope that this clarifies the distinctions.
>
> **[Discrepancy in downstream task setup: only use $X_1$ to predict $Y$ instead of $(X_1, X_2)$]**
> Our result could be extended to a representation on the full input $X=[X_1,X_2]$ by concatenating $\mathbb{E}[X_2|X_1]$ and $\mathbb{E}[X_1|X_2]$ and learning a linear layer on top of it. In the linear case, this could approximate the best linear predictor $\mathbb{E}^{L}[Y|[X_1,X_2]]$ (better than $\mathbb{E}^{L}[Y|X_1]$) with the same sample complexity benefits.
> Furthermore for the general case, under mild assumptions about the latent classes being disjoint enough, we can also show that $\mathbb{E}[Y|X_1]\approx \mathbb{E}[Y|X_1,X_2]$, suggesting that it suffices to learn $\mathbb{E}[Y|X_1]$. We will add this to the revision. In general, it is difficult to say anything mathematically precise about how a neural network trained only on part of inputs (masked inputs) will behave on the full inputs (unmasked inputs), thus making the study of the exact practical setting hard.
>
>
> **[Framework does not incorporate most existing SSL paradigms. Is it extendable to other pretext tasks like rotation? Statements should be significantly toned down]**
> We respectfully disagree with the reviewer that we have over-claimed our results. We have emphasized in our paper that we only study SSL methods that are based on reconstructing part of an input ($X_2$) from the rest of it ($X_1$), as also suggested by our title "*Predicting* what you already know helps". Our result establishes some intuitive connections between $X_1$, $X_2$ and the downstream label $Y$ that can make such SSL methods successful for downstream tasks and provides a precise mathematical formulation of this intuition. The second part of our title "Provable Self-Supervised Learning" implies provable guarantees for *one* SSL method. Indeed it would be misleading (and ever over-ambitious) to claim provable guarantees for all SSL methods and we do not intend to do so. We will modify any ambiguous statements in the paper that can be interpreted to mean otherwise.
>
> Practical examples of this "reconstruction" paradigm include image inpainting, colorization, BERT-like masked language modeling. While inpainting or colorization might not be the most powerful pretext tasks in computer vision anymore, predicting missing words is one of the most popular SSL pretext tasks in NLP. Other rotation/jigsaw-based SSL methods (predicting the transformation applied to an image), though similar to the reconstruction-based setup we consider, likely do not (or very weakly) satisfy ACI. Thus our analysis will not give meaningful bounds for these tasks, and we do not claim to do so either.
>
> As discussed in our conclusion section, our results suggest ACI as *a* mechanism by which SSL is beneficial, but do not exclude other explanations. Our work provides a thorough analysis for an SSL paradigm not analyzed before under much weaker assumptions than prior work. Understanding other SSL methods is an important future direction and we believe insights from this work can be exploited for the same.
>
> **[Comparison to [Bansal et al 2020]]**
> Their result focuses on generalization gap instead of generalization error. Their focus is not to explain whether SSL is able to find a good representation, or whether the training error is low. Instead, they show that the test error will be close to the train error, while we show *both* train and test error are small under different and non-comparable assumptions. We will add this in the revised version. Note that their paper was accessible online only after the ICLR submission deadline.

---

> > ### Comment · AnonReviewer5 · 2020-11-23
> > **Discrepancy in downstream task setup.**
> >
> > **Discrepancy in downstream task setup**
> > >  In general, it is difficult to say anything mathematically precise about how a neural network trained only on part of inputs (masked inputs) will behave on the full inputs (unmasked inputs), thus making the study of the exact practical setting hard.
> >
> > This is exactly why I am not convinced by the mathematical framework authors consider. I cannot really think of *any* self-supervised learning algorithm that does this (even the inpainting or colorization is not about this), and yet authors are claiming that the paper is about the self-supervised learning. Could authors explain more concretely why the discrepancy is not important, and the provided theory is actually meaningful?

---

> > > ### Author Response · Authors · 2020-11-24
> > > **Further clarifications**
> > >
> > > Thank you for the question. We provide some further clarifications here.
> > >
> > > In the response above, we have indicated that our results can be generalized to the case where the entire input $X=(X_1,X_2)$ is used to predict the label, by concatenating the representation we get from both $X_1$ and $X_2$. Furthermore, we can also show that in many cases, predicting the label using only un-cropped part of the image ($X_1$) is provably as good as using the entire image $(X_1,X_2)$. **In all cases, using the entire input would only make the excess risk a bit smaller than what we have proved, so our results are stronger in that sense.**
> > >
> > > **[yet authors are claiming that the paper is about self-supervised learning]**
> > > There is no doubt that our algorithm is self-supervised learning since we first learn a representation using just unlabeled data through predicting parts of inputs using the rest of the inputs. As described earlier, this framework is *motivated by* existing reconstruction-based SSL algorithms. Furthermore, in the Appendix, we conduct experiments (simulations and CV task) with the exact SSL method from our theory and find it to be effective and better than many baselines.
> > >
> > > To summarize, we can **theoretically** show that this *discrepancy* will not affect the performance by much and **experimentally** test our SSL method and already find it to be effective.
> > >
> > >
> > >
> > >
> > > It may not be possible to analyze the exact setting that is used in practice since a lot of the design choices do not make mathematical sense given our current (very little) understanding of deep learning. Thus theoretical work often ends up looking at modified versions of practical algorithms that are more amenable to precise mathematical analysis. We do the same, providing theoretical and experimental evidence that the modification is harmless. What is important here is the fresh insights that such mathematical analyses can provide, which in this case is that of approximate conditional independence.

---

### Decision · Program_Chairs · 2021-01-07
**Final Decision**

**Decision:**

Reject

**Comment:**

This paper proposes a mathematical framework to theoretically understand and quantify the benefit of self-supervision on the downstream tasks. The theoretical analyses in this paper are concrete and the authors conducted experiments to support their claims. However, the current version still has the following weaknesses.

- This paper would benefit from incorporating the reviewers' comments (which was complained by multiple reviewers) and the authors' responses if any.
- The authors need to make it clear in the abstract and introduction that (1) this paper only considers the *reconstruction-based* SSL, instead of the general SSL, and (2) addresses the discrepancy between the practice and the proposed framework.
- In the post-rebuttal phase, Reviewer 2 pointed out "Lemma 3 seems much more meaningful after the clarifications. I also notice that R5 concerned about the discrepancy in downstream task setup and may doubt the mathematical framework in this paper. In fact, I agree with R5. There is indeed a gap between the proposed mathematical model and the practical SSL algorithms. There is still some work need the authors to complete, i.e., $\mathbb{E}[Y|X_1]\approx \mathbb{E}[Y|X_1,X_2]$."